# RouteJudge: Benchmarking LLM-as-a-Judge with Routing Strategies

## Abstract

Large language models (LLMs) and large reasoning models (LRMs) are increasingly adopted as automated judges for pairwise evaluation of model outputs. However, their deployment faces three unresolved challenges: inconsistent reliability, high latency and token costs, and the lack of principled routing strategies. We introduce **RouteJudge**, the first unified framework for benchmarking and routing automated judges under accuracy–latency–cost trade-offs. Our contributions are threefold. (1) We construct six difficulty-aware datasets spanning reasoning (Math, Logic, Code) and non-reasoning (Knowledge, Roleplay, Writing) tasks, with human-verified gold standards. (2) We present the first benchmark of LRM-as-a-Judge, analyzing how intermediate thinking traces interact with final verdicts and uncovering systematic mismatches such as "good thinking but wrong verdict." (3) We develop and evaluate both offline and online routing strategies that adaptively assign judges per instance, achieving strong accuracy–efficiency trade-offs. Experiments on 19 models show that LRMs improve reasoning accuracy at higher cost, while difficulty-aware online routing narrows this gap substantially. By unifying benchmarking and routing, RouteJudge establishes the first comprehensive framework for scalable and interpretable evaluation, positioning automated judges as a practical alternative to human experts.

## 1 Introduction

Large language models (LLMs) have rapidly advanced the frontier of artificial intelligence, enabling applications in reasoning, dialogue, programming, and creative writing (OpenAI (2023b); Touvron et al. (2023); Team (2023b)). As these models are increasingly integrated into real-world systems, the demand for scalable, reliable, and cost-efficient evaluation has become pressing. Pairwise evaluation (i.e., comparing two candidate responses) is widely regarded as a more faithful reflection of human preferences than pointwise scoring (Zhou et al. (2023); Wang et al. (2023)). However, manual evaluation is both costly and difficult to scale, motivating the adoption of automated judges: either LLMs or large reasoning models (LRMs) equipped with explicit intermediate thinking.

Despite the promise of automated judging, substantial challenges remain. LLM-based judges suffer from systematic biases such as position bias, verbosity preference, and self-alignment tendencies (Dubois et al. (2023); Chen et al. (2024b)), while also exhibiting instability across repeated trials. Moreover, evaluating with stronger models introduces unavoidable accuracy–latency–cost trade-offs. Recent LRMs, designed to produce explicit and verifiable intermediate reasoning traces (DeepSeek-AI (2024)), offer new opportunities for understanding how judges reason, yet there is no prior benchmark dedicated to **LRM-as-a-Judge**, nor an analysis of how reasoning traces relate to final verdicts. Furthermore, while routing strategies have been explored for generation tasks Hao et al. (2024); Shen et al. (2023), the problem of *routing judges*—selecting among heterogeneous judges under budget constraints—remains essentially unexplored. These gaps motivate our central research question: ***How can we systematically effectively route LLMs and LRMs as automated judges, balancing accuracy, latency, and cost while remaining faithful to human preferences?***

To address these challenges, we introduce **RouteJudge**, a unified framework for benchmarking and routing automated judges. A key insight behind our design is that benchmarking and routing are intrinsically linked: effective routing requires detailed knowledge of each judge's strengths, weaknesses, reasoning behavior, and accuracy–latency–cost profile across domains and difficulty levels,

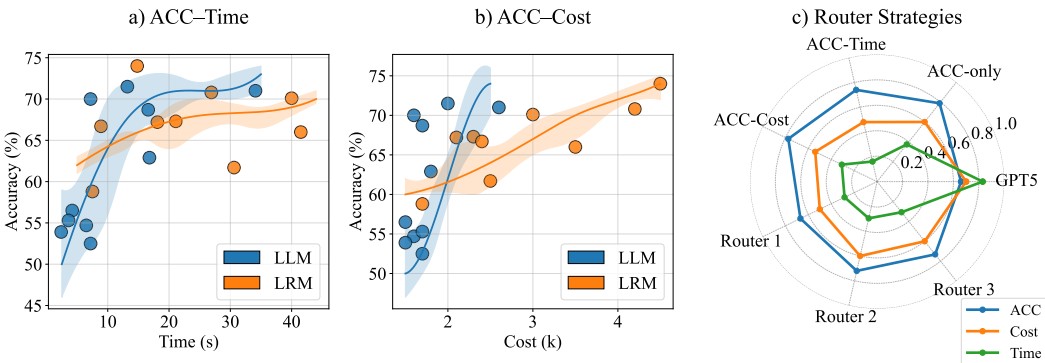

Figure 1: Representative results from RouteJudge: a) Accuracy–time trade-offs of LLM vs. LRM judges, where solid curves depict accuracy–time distributions and shaded bands indicate density. b) Accuracy–cost trade-offs. c) Efficiency gains from routing strategies under budget constraints.

while benchmarking becomes substantially more actionable when its outcomes directly guide cost-aware decision-making. RouteJudge therefore integrates both components—providing a structured benchmark that yields reliable utility signals and a routing module that operationalizes these signals into practical deployment policies. Our framework evaluates both LLM- and LRM-based judges across accuracy, latency, and cost, and incorporates routing strategies that leverage their complementary strengths. As shown in Figure 1, LLM judges are generally faster and cheaper but exhibit higher variability, whereas LRMs offer more stable and interpretable judgments at greater computational cost. Routing adaptively balances these trade-offs and achieves higher efficiency without compromising reliability. Our main contributions are as follows:

- **Benchmarking LRM-as-a-Judge.** We construct six difficulty-aware datasets spanning reasoning (Math, Logic, Code) and non-reasoning (Knowledge, Roleplay, Writing) tasks, and provide the first systematic benchmark for LRM judges against human gold standards.

- **Thinking–Verdict Analysis.** We introduce a novel analysis of LRM thinking traces, including confusion matrices between reasoning quality and verdict correctness, and an error-type taxonomy that exposes mismatches such as "good thinking but wrong verdict."

- **Routing Framework.** We design and evaluate both offline and online routing strategies that optimize accuracy–latency–cost trade-offs, showing that adaptive routing improves efficiency without sacrificing reliability.

## 2 ROUTEJUDGE

We present **RouteJudge**, a unified framework composed of three modules as shown in Fig. 2.

### 2.1 PRELIMINARIES AND PROBLEM SETUP

Each evaluation instance $x \in \mathcal{X}$ is paired with two free-form candidate responses $(A, B)$ produced by distinct generators.[1] The judge must select a preference from $\mathcal{Y} = \{A, B, \text{Tie}\}$.

**Judge the hypothesis class.** A judge model $m$ implements a three-way probabilistic classifier

$$\mathbf{p}_m(x; A, B) = \left(p_m^{\mathrm{A}}, p_m^{\mathrm{B}}, p_m^{\mathrm{Tie}}\right) \in \Delta^2, \tag{1}$$

and outputs a verdict

$$\hat{y}_m(x) = \arg\max_{y \in \mathcal{Y}} p_m^y(x; A, B), \qquad c_m(x) = \max_y p_m^y(x; A, B). \tag{2}$$

**Unified LLM/LRM interface.** LLMs return verdict-only outputs, while LRMs additionally produce explicit thinking traces. Throughout the paper we use "thinking trace" as a unified term referring to any model-generated intermediate reasoning. This consistent API enables direct comparison between LLMs and LRMs and supports analysis of how intermediate reasoning relates to final verdicts. Thinking traces provide a structured diagnostic signal for understanding judge behavior.

---

[1]We use GPT-4o and Claude 3.5 as the two response generators for all domains.

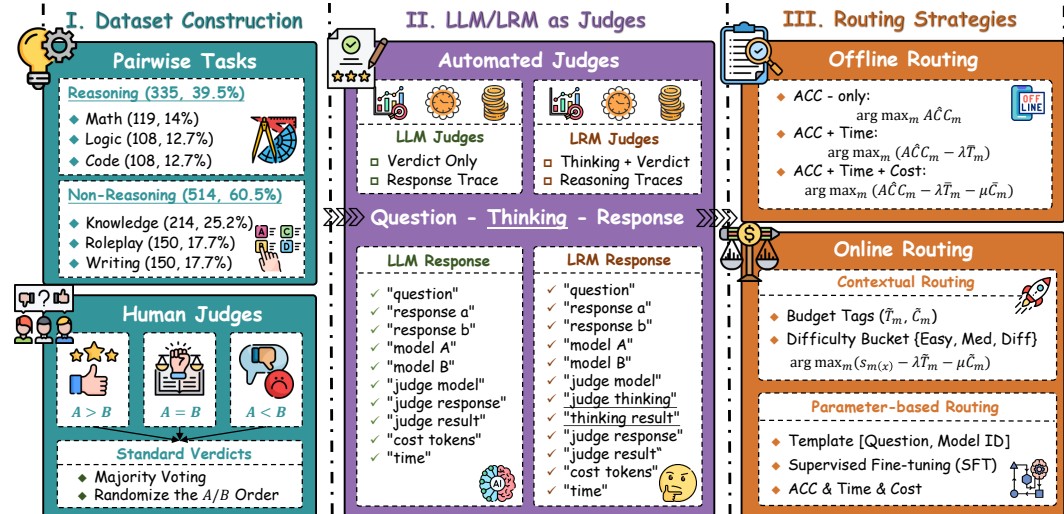

Figure 2: RouteJudge pipeline. Stage I: dataset construction with pairwise tasks and human judges. Stage II: judge pool with a unified interface; LRMs additionally produce thinking traces for interpretability. Stage III: routing strategies, including offline rankers and online routers.

## 2.2 DATASET CONSTRUCTION

**Pairwise task formation.** We curate two super-domains to probe complementary capabilities: reasoning (math, logic, code) and non-reasoning (knowledge, roleplay, writing). For each instance $x$, we obtain two responses $(A, B)$ from distinct generators. To mitigate position bias we uniformly randomize the order of $(A, B)$, and we log auxiliary features for later analysis. We additionally provide both standard and thinking variants (including explicit reasoning traces from the generators).

**Standard human verdicts.** Each triplet $(x, A, B)$ is annotated by $n = 7$ independent raters with labels $z_i(x) \in \mathcal{Y}$. The majority-vote label $y^{\mathrm{MV}}(x) = \arg\max_{y \in \mathcal{Y}} \sum_{i=1}^{n} \mathbf{1}[z_i(x) = y]$ serves as the gold standard for evaluation. Inter-rater agreement and tie statistics are provided in Appendix D.

## 2.3 LRM-AS-A-JUDGE BENCHMARK

We extend the judge to LRMs with {*verdict, confidence, rationale, thinking*}. Unlike conventional LLMs, LRMs produce both verdicts and intermediate reasoning traces. This dual output opens the door to new questions: *Does high-quality reasoning necessarily yield correct verdicts? Can flawed reasoning still produce superficially correct answers?* By benchmarking LRMs across reasoning and non-reasoning domains, we establish the first large-scale evaluation of judge models that reason explicitly. This benchmark not only measures verdict accuracy but also highlights where thinking traces add interpretability, offering a diagnostic lens absent from prior LLM-only evaluations.

## 2.4 ROUTING STRATEGIES

Given an instance $x$, the routing module must decide which judge $m \in \mathcal{M}$ to invoke. For each judge, we track its expected accuracy, latency, and token cost, and define the utility

$$U(m \,|\, x) \;=\; \underbrace{\mathbb{E}[\mathrm{ACC}_m(x)]}_{\text{accuracy}} \;-\; \lambda \underbrace{\mathbb{E}[T_m(x)]}_{\text{time}} \;-\; \mu \underbrace{\mathbb{E}[C_m(x)]}_{\text{token cost}}, \tag{3}$$

with application-specific weights $\lambda, \mu \geq 0$. We also allow an abstention action $\perp$ (fallback-to-human) with utility $U(\perp \,|\, x)$. Figure 3 illustrates the overall routing framework: offline policies optimize global averages, while online strategies adaptively route per instance.

**Offline Routing (Instance-agnostic).** Using a domain-balanced validation split, we compute global accuracy $\hat{\mathrm{ACC}}_m$, mean latency $\overline{T}_m$, and mean token cost $\overline{C}_m$ for all judges. These statistics define three fixed policies: (1) an accuracy-only policy: $\pi_{\mathrm{A}}(x) = \arg\max_m \hat{\mathrm{ACC}}_m$, (2) an accuracy–time policy: $\pi_{\mathrm{AT}}(x) = \arg\max_m (\hat{\mathrm{ACC}}_m - \lambda \overline{T}_m)$, (3) an accuracy–time–cost:

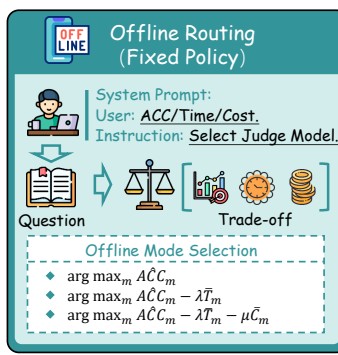 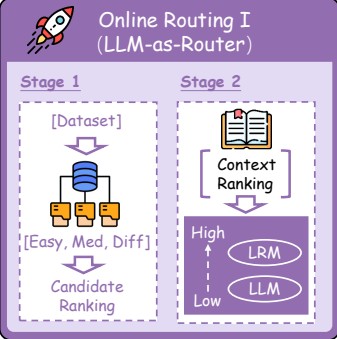 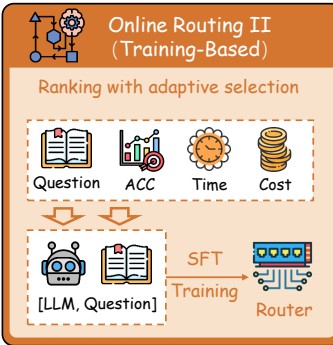

Figure 3: **Routing framework overview.** Offline strategies rely on global statistics to rank models, producing fixed policies (ACC-only, ACC-Time, ACC-Time-Cost). Online strategies adapt at the instance level: (i) Contextual routing (LLM-as-Router) leverages structured prompts and difficulty-aware buckets; (ii) Training-based routing trains a lightweight router on inexpensive features.

$\pi_{\text{ATC}}(x) = \arg\max_m(\hat{\text{ACC}}_m - \lambda \overline{T}_m - \mu \overline{C}_m)$. Although easy to deploy and fully reproducible, these policies cannot incorporate instance-level variation such as domain, difficulty, or input complexity, and therefore serve as strong baselines for evaluating adaptive routing.

**Online Routing I: Contextual (LLM-as-Router).** To enable adaptivity, we employ an LLM router. The router is prompted with compact features of $x$ (length, domain cues), a brief legend of each judge's historical strengths, and budget tags $(\tilde{T}_m, \tilde{C}_m)$. In our difficulty-aware variant, we first map $x$ to a difficulty bucket $b \in \{\text{Easy}, \text{Medium}, \text{Diff}\}$ using lightweight heuristics; we then restrict attention to a pre-defined model band $\mathcal{S}_b$ constructed from offline cost-aware rankings (low $\to$ high within the band). The router outputs scores $\{s_m(x)\}_{m \in \mathcal{S}_b}$ and we select

$$\pi_{\text{ctx}}(x) = \arg\max_{m \in \mathcal{S}_b} \left(s_m(x) - \lambda \tilde{T}_m - \mu \tilde{C}_m\right). \tag{4}$$

A confidence gate $\max_m s_m(x) < \tau$ triggers escalation within the band (low$\to$high) or abstention $\perp$. This design couples global efficiency with local adaptivity, yielding robust budget adherence.

**Online Routing II: Training-based (Trainable Router).** We additionally develop a training-based router that learns routing decisions from data. The router is a lightweight 7B–8B model $r_\theta : \phi(x) \to \Delta^{|\mathcal{M}|+1}$, where $\phi(x)$ contains inexpensive input features such as the question text and the judge recommended by the offline utility-maximizing policy. Gold labels, difficulty annotations, and human preference signals are never used as router inputs, ensuring that routing decisions are not driven by label-derived information. Training data are constructed by pairing each instance $x$ with its offline-selected judge, treated as the supervisory target. The router is trained to predict a distribution over all candidate judges that aligns with utility-maximizing choices, allowing it to override the offline policy when the input features suggest a potential improvement in accuracy, time, or cost. The dataset is split into train, validation, and test by domain to prevent leakage of instance-level signals across splits. At inference time, the router processes a short input prompt and outputs a single routing decision, adding negligible overhead relative to invoking an LLM or LRM judge. The training-based router therefore provides a scalable, low-cost mechanism for incorporating instance-specific information into routing policies while maintaining clear separation between routing features and human evaluation signals.

Table 1: Overview of the **RouteJudge**.

| Category | Value |
|---|---|
| **Corpus** | |
| Total items | 849 |
| Reasoning : Non-reasoning | 335 : 514 |
| Candidate responses / item | 2 (A,B) |
| Total judgments | 5943 |
| **Domain distribution** | |
| Validation : Test | 96 : 753 |
| Math | 119 (14%) |
| Logic | 108 (12.7%) |
| Code | 108 (12.7%) |
| Knowledge | 214 (25.2%) |
| Roleplay | 150 (17.7%) |
| Writing | 150 (17.7%) |
| **Difficulty** | |
| Easy : Easy-Think | 323 : 239 |
| Medium : Medium-Think | 165 : 121 |
| Diff : Diff-Think | 175 : 83 |

## 3 EXPERIMENT

### 3.1 EXPERIMENT SETUP

**Dataset.** The **RouteJudge** corpus supports systematic evaluation of automated judges under a unified pairwise setting. Each instance contains a prompt and two candidate responses $(A, B)$ generated by **GPT-4o** and **Claude 3.5**, following JudgeBench and FairEval to ensure stylistic diversity and avoid single-model bias. Human gold labels come from seven independent annotators. The dataset spans six domains—Math, Logic, Code, Knowledge, Roleplay, Writing—grouped into reasoning and non-reasoning. Difficulty buckets (EASY/MEDIUM/DIFF) are assigned by evaluating each item across several baseline judges and partitioning by average correctness, producing a model-agnostic difficulty split. For each item, we also include a THINK variant in which the original generators provide explicit reasoning traces. These paired versions allow controlled comparison of verdict-only judging (LLMs) and thinking-augmented judging (LRMs). Full data provenance and domain composition are provided in Appendix A.1.

**Models.** We evaluate 19 judge models spanning both LLMs and LRMs, covering a broad spectrum of accuracy–efficiency trade-offs. Ten models are standard LLM judges that output a direct verdict without structured intermediate reasoning. Nine models are LRMs (e.g., DeepSeek-R1, o3-mini) that generate explicit thinking traces before producing a final judgment. LRMs and LLMs are handled through a unified judge interface. To ensure comparability across models, all LLM judges are evaluated with temperature $= 0$ for deterministic verdicts. LRM judges are evaluated using their recommended decoding temperatures to preserve reasoning stability, while all models share a global maximum output length of 1024 tokens for consistent latency and cost measurement.

### 3.2 MAIN RESULTS

Table 3 reports accuracy across reasoning and non-reasoning domains. To contextualize variability in human preference, we compute inter-annotator agreement on the RouteJudge corpus (Table 2). Structured tasks such as Math and Logic yield higher agreement (Fleiss' $\kappa$ between 0.78–0.81), whereas Roleplay and Writing show substantially more near-ties (21–23%), reflecting their inherent subjectivity. Three observations emerge. First, LRMs obtain the strongest performance in reasoning domains: *Gemini-2.5-Flash* (74.90%) and *Gemini-2.5-Pro* (73.63%) lead in Math and Logic. Paired bootstrap tests (10k samples) yield narrow confidence intervals (average width 1.7 pp), confirming statistical robustness.

Table 2: Inter-annotator agreement, tie/near-tie rates, and difficulty statistics. Agreement uses Krippendorff's $\alpha$ and Fleiss' $\kappa$. Near-ties denote 4–3 or 3–3–1.

| **Domain** | $\alpha$ | $\kappa$ | Tie / Near-tie |
|---|---|---|---|
| Coding | 0.79 | 0.76 | 14.8% |
| Math | 0.83 | 0.81 | 11.2% |
| Logic | 0.82 | 0.78 | 12.6% |
| Knowledge | 0.76 | 0.72 | 18.4% |
| Roleplay | 0.74 | 0.70 | 21.7% |
| Writing | 0.71 | 0.68 | 23.3% |
| Overall | 0.78 | 0.74 | 17.0% |

Second, in non-reasoning domains, compact instruction-following models remain competitive. *Claude-3.5-Sonnet* achieves the best Writing and Roleplay accuracy, highlighting that explicit multi-step reasoning is less beneficial when human preference is driven by tone, pragmatics, or stylistic alignment. Third, domain specialization is evident: some models (e.g., *Gemma-3-27b-it*) excel in Coding and Logic but lag in subjective domains, whereas others (e.g., *o3-Mini*) score well in Logic but struggle in Math. Intermediate models such as *Qwen3-32B* and *DeepSeek-R1* show the lowest cross-domain variance, suggesting architectural robustness rather than scale alone improves generality. We additionally clarify that "Avg. Thinking Length" in Table 1 is measured in tokens (via GPT-4o tokenizer). Explicit reasoning length correlates positively with accuracy on reasoning tasks ($r = 0.28$) but shows negligible correlation on subjective tasks. Finally, temperature-sensitivity experiments indicate that standard LLMs behave most stably at temperature 0, whereas LRMs such as *DeepSeek-R1* and *Gemini-2.5-Flash* perform slightly better at 0.5–0.6 (typically +1–2 pp). We therefore use recommended settings for each family in all evaluations.

### 3.3 TIME AND COST LATENCY RESULTS

Figure 4 summarizes the accuracy–latency–cost trade-offs. LLMs form a low-cost, low-latency frontier but do not achieve the highest accuracy. LRMs incur substantially more tokens because of explicit reasoning traces; for example, *Gemini-2.5-Flash* produces longer thinking sequences (me-

Table 3: Comparison of judge accuracy (%) across reasoning and non-reasoning tasks. Best results per column are in **bold**, second-best are underlined. Within each row, the maximum and minimum domain scores are highlighted in green and red, respectively.

| Model | Split | | Reasoning | | | | Non-Reasoning | | | | Overall |
|---|---|---|---|---|---|---|---|---|---|---|---|
| | Val | Test | Coding | Math | Logic | Avg | Knowledge | Roleplay | Writing | Avg | |
| **Baselines** | | | | | | | | | | | |
| Random Choice | 33.3 | 33.3 | 33.3 | 36.97 | 36.11 | 35.46 | 33.18 | 36.67 | 33.3 | 34.39 | 34.93 |
| Frequent Choice | 36.82 | 35.91 | 38.03 | 37.53 | 36.27 | 37.23 | 34.54 | 36.71 | 34.95 | 35.41 | 36.37 |
| Expert (Human & GPT4o) | 83.24 | 77.93 | - | - | - | - | - | - | - | - | - |
| **Large Language Models (without Thinking)** | | | | | | | | | | | |
| Llama-3.1-8B-Instruct | 52.08 | 49.34 | 48.15 | 46.22 | 59.26 | 51.21 | 57.48 | 52.00 | 65.33 | 58.27 | 54.74 |
| Llama-3.3-70B-Instruct | 59.38 | 56.17 | 57.41 | 55.46 | 63.89 | 58.92 | 65.42 | 60.67 | 74.67 | 66.92 | 62.25 |
| Qwen2.5-7B-Instruct | 62.50 | 58.96 | 57.41 | 49.58 | 60.19 | 55.73 | 56.07 | 52.00 | 64.00 | 57.36 | 56.88 |
| Qwen2.5-72B-Instruct | 65.62 | 62.33 | 69.44 | 70.59 | 67.59 | 69.21 | 69.63 | 61.33 | 73.33 | 68.10 | 68.65 |
| Qwen3-32B | 70.83 | 67.44 | 70.37 | 73.95 | 72.22 | 72.18 | 71.03 | 65.33 | 73.33 | 69.90 | 71.04 |
| Gemma-3-4b-it | 53.12 | 51.07 | 49.07 | 49.58 | 58.33 | 52.33 | 54.67 | 56.00 | 64.00 | 58.22 | 55.61 |
| Gemma-3-27b-it | 68.75 | 65.92 | 75.00 | 64.71 | 75.00 | 71.57 | 68.22 | 64.00 | 73.33 | 68.52 | 70.38 |
| GPT-3.5-Turbo-1106 | 54.17 | 51.68 | 49.07 | 58.82 | 49.07 | 52.32 | 50.47 | 48.00 | 68.00 | 55.49 | 53.57 |
| GPT4o | 45.83 | 43.11 | 51.85 | 52.10 | 50.93 | 51.63 | 52.80 | 48.00 | 59.33 | 53.38 | 52.50 |
| Claude-3.5-Sonnet | 71.88 | 68.72 | 73.14 | 67.23 | 70.37 | 70.25 | 73.83 | 71.33 | 73.33 | 72.83 | 71.71 |
| **Large Reasoning Models (with Thinking)** | | | | | | | | | | | |
| QwQ-32B | 73.96 | 70.43 | 70.37 | 70.59 | 68.52 | 69.83 | 70.56 | 66.67 | 74.00 | 70.41 | 70.45 |
| Kimi-K2-Instruct | 56.25 | 53.77 | 62.04 | 47.90 | 57.41 | 55.78 | 61.68 | 60.00 | 64.00 | 61.89 | 58.84 |
| Doubao-1.5-Thinking-Pro | 63.54 | 59.93 | 59.26 | 46.22 | 58.33 | 54.60 | 65.89 | 64.00 | 76.67 | 68.85 | 61.06 |
| DeepSeek-R1 | 70.83 | 67.08 | 73.15 | 59.66 | 65.74 | 66.18 | 69.63 | 63.33 | 72.67 | 68.54 | 67.03 |
| Claude-3.7-Sonnet | 68.75 | 65.62 | 66.67 | 64.71 | 64.81 | 65.40 | 72.43 | 66.00 | 68.67 | 69.03 | 67.55 |
| Gemini-2.5-Flash | 69.79 | 66.45 | 75.00 | 75.63 | 74.07 | 74.90 | 72.90 | 67.33 | 78.00 | 72.74 | 73.82 |
| Gemini-2.5-Pro | 66.04 | 63.25 | 73.15 | 80.00 | 67.74 | 73.63 | 67.44 | 63.33 | 73.33 | 68.03 | 70.83 |
| o3-Mini | 63.54 | 60.18 | 71.30 | 56.30 | 75.00 | 67.53 | 67.76 | 58.00 | 72.00 | 65.92 | 66.73 |
| GPT5 | 66.67 | 63.82 | 62.96 | 61.34 | 73.15 | 65.82 | 68.69 | 64.00 | 66.00 | 66.23 | 66.69 |

Figure 4: Radar analysis of accuracy–efficiency trade-offs across judges. For readability, model names are abbreviated as follows: Cl → Claude, Lm → Llama, Q → Qwen, Gm → Gemini, Gem → Gemma, DS → DeepSeek, Db → Doubao. a) Accuracy distribution across six domains, b) average latency (s), and c) normalized token cost.

dian 250 tokens), explaining its higher cost despite a 1024-token cap. Mid-sized models (*Qwen3-32B*, *DeepSeek-R1*) trace out a Pareto-efficient region, reducing latency by 35–50% relative to the strongest LRMs while staying within 2–3 pp of their accuracy. Variance profiles also differ: LLMs exhibit tight latency distributions, whereas multi-stage LRM reasoning induces broader spreads. These patterns further motivate routing mechanisms for accuracy–efficiency balancing.

## 3.4 PIPELINE DEMONSTRATION

Figure 5 illustrates how RouteJudge integrates human annotation, automated judging, and routing into a coherent workflow. The pipeline is designed around two principles. First, a unified judge interface enables LLMs and LRMs to be evaluated under identical pairwise settings, while allowing LRMs to contribute structured thinking traces for interpretability. Second, routing is naturally layered: offline policies provide cost-aware global defaults, whereas online routing introduces

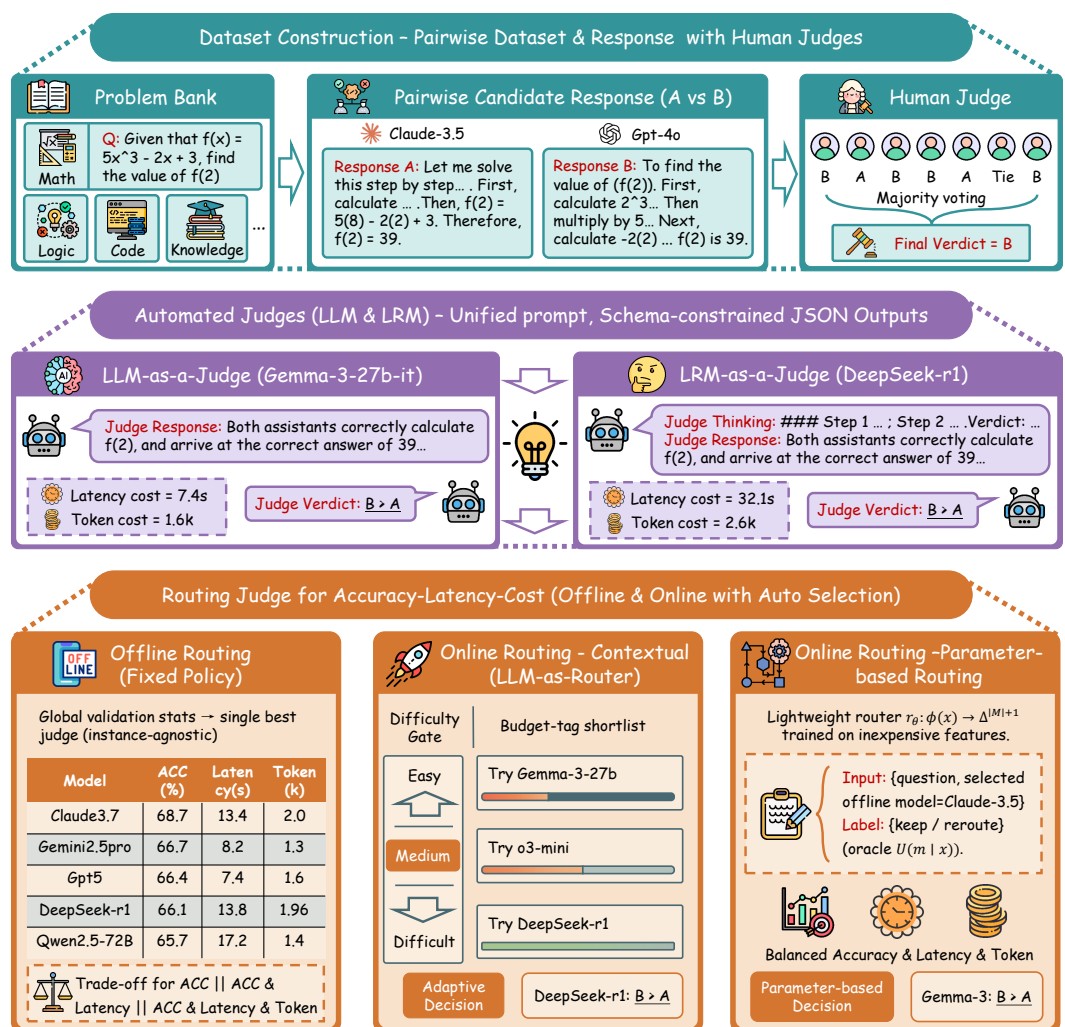

Figure 5: RouteJudge pipeline overview: dataset construction with human-preferred labels, automated judges producing verdicts (LLMs) or verdicts plus thinking traces (LRMs), and routing modules balancing accuracy, latency, and cost.

instance-level adaptivity through contextual prompting or lightweight learned routers. This structure mirrors real-world deployment considerations, where judgments must remain reliable under budget constraints while leveraging explicit reasoning when beneficial.

## 3.5 ROUTING STRATEGY COMPARISON

Table 4 compares fixed-model baselines with offline and online routing strategies. Offline heuristics provide strong global references: ACC-only maximizes accuracy by always selecting the strongest judge, ACC–Time minimizes latency by prioritizing compact LLMs, and ACC–Time–Cost trades off accuracy against both time and tokens. However, these policies ignore instance-level variation in domain and difficulty. The online routers address this limitation. Router 2 uses *Llama-3.1-8B* as a backbone and Router 3 uses *Qwen2.5-7B*. These models were selected because they are substantially lighter than the strongest judges (30–70B or proprietary LRMs) yet retain enough capacity to detect structural patterns in prompts and candidate responses. Router 2 tends to prefer judges that perform well on subjective tasks (e.g., Knowledge, Writing), reflecting the backbone's strength in fluency and stylistic cues. Router 3 performs best in Coding and Math, domains where shallow heuristics are insufficient and structural features (e.g., problem format, numerical operators) reliably map to stronger LRMs. This specialization yields distinct benefits. Router 2 achieves the highest non-reasoning accuracy (e.g., 77.2% in Writing), whereas Router 3 achieves the highest accuracy in

Table 4: Quantitative comparison of routing strategies across datasets. Best accuracy per column is highlighted in green , best efficiency (lowest Time/Cost) in red .

| Strategy | Coding | | | Knowledge | | | Math | | | Reasoning | | | Roleplay | | | Writing | | | Average | | |
|---|---|---|---|---|---|---|---|---|---|---|---|---|---|---|---|---|---|---|---|---|---|
| | ACC | T | C | ACC | T | C | ACC | T | C | ACC | T | C | ACC | T | C | ACC | T | C | ACC | T | C |
| **Baseline** | | | | | | | | | | | | | | | | | | | | | |
| GPT5 (fixed) | 62.9 | 44.8 | 4.7 | 68.7 | 23.2 | 2.8 | 61.3 | 19.5 | 2.7 | 73.2 | 77.2 | 5.1 | 64.0 | 35.0 | 2.4 | 66.0 | 49.1 | 3.1 | 66.0 | 41.5 | 3.5 |
| **Offline Routing** | | | | | | | | | | | | | | | | | | | | | |
| ACC-only | 94.4 | 37.4 | 4.1 | 73.8 | 13.9 | 1.9 | 80.0 | 25.5 | 4.3 | 75.0 | 12.3 | 3.0 | 71.3 | 13.2 | 1.7 | 78.0 | 10.0 | 3.1 | 78.8 | 18.7 | 3.0 |
| ACC–Time | 75.0 | 8.3 | 2.1 | 72.9 | 6.2 | 2.3 | 75.6 | 8.9 | 3.0 | 75.0 | 7.0 | 1.6 | 67.3 | 8.2 | 2.3 | 78.0 | 10.0 | 3.1 | 74.0 | 8.1 | 2.4 |
| ACC–Time–Cost | 92.0 | 35.5 | 3.7 | 72.5 | 12.1 | 1.8 | 78.3 | 21.2 | 4.0 | 74.5 | 11.0 | 2.0 | 70.0 | 12.7 | 1.6 | 77.5 | 9.8 | 2.5 | 77.5 | 15.4 | 2.7 |
| **Online Routing** | | | | | | | | | | | | | | | | | | | | | |
| Router 1 | 68.8 | 8.8 | 2.7 | 56.3 | 12.6 | 1.7 | 68.6 | 29.2 | 5.3 | 70.0 | 12.4 | 2.3 | 62.5 | 13.5 | 1.7 | 75.0 | 8.4 | 1.2 | 66.9 | 14.2 | 2.5 |
| Router 2 (Llama) | 72.5 | 11.4 | 3.4 | 70.8 | 14.8 | 2.5 | 71.0 | 20.6 | 4.2 | 72.0 | 14.9 | 2.8 | 68.0 | 15.2 | 2.4 | 77.2 | 12.1 | 2.0 | 72.0 | 14.8 | 3.0 |
| Router 3 (Qwen) | 74.6 | 12.8 | 3.8 | 69.1 | 15.6 | 2.6 | 76.3 | 19.4 | 4.0 | 73.5 | 16.2 | 2.9 | 69.4 | 16.5 | 2.2 | 76.0 | 11.9 | 2.1 | 73.2 | 15.4 | 3.0 |

a) Thinking Error Distribution    b) Response Error Distribution    c) Confusion Heatmap

Figure 6: Analysis of reasoning-level and verdict-level errors. Left: distribution of thinking-trace errors. Middle: response-level evaluation errors. Right: confusion heatmaps showing alignment or mismatch between reasoning quality (TR/TW) and final verdict correctness (JR/JW).

Coding and Math (74.6%, 76.3%). Both routers reduce latency by 20–45% relative to ACC-only, and bootstrap tests indicate that Router 3 significantly exceeds ACC–Time in Coding and Math (p ¡ 0.05). Router 1, which relies only on template-level cues, is efficient but insufficiently accurate. **Why Train a Router?** While LLM-as-Router enables contextual routing, it remains relatively expensive and exhibits instability across domains and prompt variants. The trainable router offers a complementary design: it is 20–30× cheaper to evaluate, requires only lightweight features, and generalizes more robustly under distribution shift. In practice, we find that learned routing achieves comparable or superior accuracy while substantially reducing overhead, making it more suitable for large-scale or latency-sensitive deployment.

## 3.6 THINKING VS. RESPONSE: ERROR-TYPE ANALYSIS

Figure 6 analyzes discrepancies between intermediate reasoning and final verdicts. We distinguish thinking (explicit multi-step traces produced by LRMs) from short rationales occasionally returned by LLMs; only the former represent structured reasoning sequences. To ensure reliability, a subset of the error labels and confusion categories was cross-checked by human annotators. Thinking-level errors are dominated by omitted details (33.2%), feature bias (30.6%), and incorrect evaluation criteria (23.7%), while logical inconsistencies are infrequent. At the response level, omissions become less common but reliance on incorrect criteria increases, indicating that verdict construction can compensate for missing steps yet still fail by misapplying evaluation rules. The confusion heatmaps reveal how reasoning quality aligns with verdict correctness. Objective domains such as Math and Logic exhibit strong TR–JR alignment (over 90%), suggesting that reasoning traces play a functional role in these tasks. In contrast, subjective domains show larger mismatches: Knowledge has a considerable share of good thinking + wrong verdict (10.4%), indicating under-utilization of sound reasoning, whereas Writing shows high flawed thinking + correct verdict (30.6%), reflecting reliance on stylistic or fluency cues rather than reasoning consistency. These patterns show that thinking traces are informative but not fully faithful: strong reasoning does not guarantee correct decisions, and flawed reasoning can still yield acceptable verdicts when models rely on surface-level cues. This finding aligns with recent concerns about the faithfulness of chain-of-thought reasoning and highlights the need for reasoning-aware calibration and routing.

Table 5: Difficulty-aware split with cross-bucket resolvability. Each row reports bucket sizes (count; proportion within category) and unresolved counts (absolute; relative to the higher bucket).

| Category | Bucket size (count; % within category) | | | Cross-bucket Unresolved (n; rate) | | | Total (share) |
|---|---|---|---|---|---|---|---|
| | Easy | Medium | Diff | Diff $\not\to$ Med | Diff $\not\to$ Easy | Med $\not\to$ Easy | |
| **Non-reasoning** | 323  (48.7%) | 165  (24.9%) | 175  (26.4%) | 41  (23.4%) | 60  (34.3%) | 52  (31.5%) | 663  (60.0%) |
| **Reasoning** | 239  (54.0%) | 121  (27.3%) | 83  (18.7%) | 31  (27.7%) | 29  (25.9%) | 52  (46.4%) | 443  (40.0%) |

## 3.7 DIFFICULTY-AWARE RESULTS

To examine how judges behave under varying task difficulty, we introduce a stratified split: non-reasoning tasks are partitioned into *Easy/Medium/Diff*, while reasoning tasks are grouped into *Easy-Think/Medium-Think/Diff-Think*. This design enables us to test whether stronger judges can resolve harder subsets that systematically defeat weaker ones. As shown in Table 5, reasoning tasks with explicit thinking traces exhibit sharper stratification than non-reasoning tasks. Nearly half of the *Medium-Think* items remain unresolved when falling back to *Easy-Think* (46.4%), and over a quarter of *Diff-Think* cases remain unsolved even when stepping down to the easiest bucket (25.9%). In contrast, non-reasoning tasks display milder separation, with unresolved rates concentrated in the 23–34% range. These findings yield two insights. First, reasoning-intensive domains naturally amplify difficulty gaps, validating the importance of difficulty-aware benchmarks for probing robustness. Second, unresolved transitions provide actionable signals for routing: high-difficulty items should be escalated to stronger (but costlier) judges, while easier subsets can be reliably delegated to cheaper ones. Consistent with this view, Figure 6 further visualizes unresolved flows across buckets, reinforcing how difficulty-aware stratification aligns with multi-objective routing utilities.

## 4 RELATED WORK

**LLMs as Judges.** LLMs have been used as automated evaluators in MT-Bench and Chatbot Arena (Zheng et al., 2023b), but exhibit systematic biases—position, verbosity, and self-preference (Chen et al., 2024a; Shi et al., 2024; Wataoka et al., 2024; 2025; Li et al., 2024). Broader analyses highlight instability and inconsistency (Zhang et al., 2025; Anghel et al., 2025), suggesting that reliable evaluation requires multiple judges and principled routing rather than a single fixed evaluator.

**Reasoning Models and Thinking Traces.** LRMs provide explicit reasoning, yet prior work shows such traces may be helpful or misleading (Anthropic, 2025a; Apple Machine Learning Research, 2025). Recent LRMs (e.g., DeepSeek-R1 (Guo et al., 2025b), o3/o3-mini (OpenAI, 2025a;b)) emphasize systematic reasoning, but no study evaluates LRM-as-a-Judge. Our benchmark addresses this via domain-wise confusion analysis and an error taxonomy for both thinking and judgments.

**Routing and Model Selection.** Routing in generation tasks has been explored through cascades (Chen et al., 2023), preference-trained routers (Ong et al., 2024a;b; Li et al., 2023), and efficient verification (Narasimhan et al., 2025). Judge routing raises additional challenges: heterogeneous biases (Trivedi et al., 2024; Sun et al., 2025; Ye & Ng, 2024), calibrated abstention (Jung et al., 2024), and reasoning-quality effects in LRMs (Chen et al., 2025). We study these factors jointly by routing across LLM and LRM judges with explicit accuracy–latency–cost considerations.

## 5 CONCLUSION

We presented **RouteJudge**, a unified framework for evaluating and routing automated judges. Our results reveal three overarching insights. First, LRMs provide more stable judgments on reasoning tasks, yet their explicit thinking incurs substantial latency and cost, showing that reasoning depth is beneficial but not universally efficient. Second, analysis of thinking–verdict mismatches indicates that correct reasoning does not always translate into correct decisions, motivating future work on reasoning-aware calibration. Third, difficulty-aware stratification exposes clear separations between easy and hard subsets, enabling routing policies that allocate weak judges to easy items while reserving strong LRMs for genuinely challenging cases. Finally, online routing consistently improves efficiency without sacrificing accuracy, approaching expert-level performance under realistic budgets. We hope RouteJudge offers a foundation for developing cost-aware, reliable, and interpretable evaluation systems as automated judging becomes increasingly central to LLM deployment.

ETHICS STATEMENT

All authors confirm adherence to the ICLR Code of Ethics throughout this work. This study does not involve human participants in sensitive or personal contexts; instead, gold-standard annotations were obtained from professional annotators with domain expertise, who labeled paired model responses under controlled guidelines. No personally identifiable or sensitive information is included in the dataset, and all model outputs originate from publicly available LLMs or LRMs. We have taken care to report methods and results transparently, avoid misrepresentation, and disclose all relevant implementation details. Potential societal impacts were considered: automated judges may influence downstream evaluation pipelines, and our analysis highlights both benefits (scalable benchmarking, efficiency-aware routing) and risks (bias amplification, miscalibration). We emphasize that RouteJudge is intended solely for research purposes, and the framework should not be deployed in high-stakes applications without additional safeguards. The authors declare no conflicts of interest.

REPRODUCIBILITY STATEMENT

To facilitate reproducibility, we release supplementary materials accompanying this submission. These include: (i) the curated RouteJudge dataset with difficulty-aware splits across reasoning and non-reasoning domains; (ii) all evaluation code for judge benchmarking and routing strategies; and (iii) detailed instructions covering environment setup, dependency requirements, and execution steps for reproducing all experiments. Key hyperparameters (e.g., decoding settings, maximum output length, cost and latency measurement protocols) are explicitly reported in the main text, while annotation aggregation rules and error-type taxonomies are documented in the supplementary materials. Together, these resources enable independent researchers to replicate our experiments, validate reported results, and extend the framework to new models or routing policies.

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

Large Language Models (LLMs) were used to aid in the writing and polishing of the manuscript. Specifically, we used an LLM to assist in refining the language, improving readability, and ensuring clarity in various sections of the paper. The model helped with tasks such as sentence rephrasing, grammar checking, and enhancing the overall flow of the text.

It is important to note that the LLM was not involved in the ideation, research methodology, or experimental design. All research concepts, ideas, and analyses were developed and conducted by the authors. The contributions of the LLM were solely focused on improving the linguistic quality of the paper, with no involvement in the scientific content or data analysis.

The authors take full responsibility for the content of the manuscript, including any text generated or polished by the LLM. We have ensured that the LLM-generated text adheres to ethical guidelines and does not contribute to plagiarism or scientific misconduct.

# A  ADDITIONAL DETAILS

## A.1  DATASET COMPOSITION

RouteJudge differs from prior judge-oriented datasets (e.g., MT-Bench, FairEval, JudgeBench) in three important ways. First, it spans six domains across both reasoning and non-reasoning tasks, forming a broader evaluation space than existing judge benchmarks, which typically focus on only two or three domains. Second, RouteJudge provides paired Think/Non-Think variants for every item, enabling controlled analyses of explicit reasoning traces—an aspect absent in earlier datasets. Third, we adopt a dual-source response-generation protocol using GPT-4o and Claude 3.5, following the convention of prior judge studies while ensuring stylistic diversity and reducing model-specific bias. These design choices make RouteJudge the first difficulty-aware, reasoning-augmented benchmark tailored specifically for studying both LLM and LRM judges.

Table 6 presents a consolidated summary of the **RouteJudge** corpus, including data provenance, domain definitions, pairwise construction, and human annotation metadata. This complements the high-level description in Section 4.1 and addresses reviewer questions regarding dataset sources, difficulty assignment, and thinking–response variants.

**Data provenance and coverage.**  RouteJudge aggregates 849 pairwise instances drawn from twelve publicly available datasets (e.g., MT-Bench, FairEval, JudgeBench, APPS, MathQA). These sources span a broad spectrum of reasoning skills (symbolic logic, mathematical derivation, code correctness) and non-reasoning capabilities (factual knowledge, roleplay consistency, stylistic writing). Compared to prior judge benchmarks (e.g., MT-Bench, JudgeBench), RouteJudge covers more domains, more difficulty diversity, and includes explicit reasoning variants, enabling multidimensional analysis.

**Pairwise construction aligned with judge literature.**  For each task, we generate two heterogeneous responses $(A, B)$ using **GPT-4o** and **Claude 3.5**, following established practice in JudgeBench and FairEval. This design ensures stylistic variety and avoids same-model correlations, yielding more realistic and challenging pairwise comparisons for automated judges.

**Difficulty-aware clustering.**  A key innovation of RouteJudge is its **model-agnostic difficulty partition**. We evaluate each instance with a panel of baseline judges and assign Easy/Medium/Diff buckets via average correctness. This yields a difficulty structure that generalizes across different judge architectures and supports routing analyses that rely on difficulty escalation (e.g., Easy $\rightarrow$ Medium $\rightarrow$ Diff).

**Thinking and non-thinking variants.**  For all items, we provide a parallel THINK version where response generators include explicit reasoning traces. This enables controlled experiments comparing verdict-only judges with LRM judges that consume intermediate thoughts. The ability to jointly evaluate verdict correctness, reasoning quality, and thinking–verdict mismatches is a distinctive contribution of RouteJudge and addresses gaps noted by reviewers (e.g., the lack of LRM-as-judge benchmarks).

**Human gold annotation and reliability.** Each instance is rated by seven independent annotators. Beyond majority vote, we record rater distributions and compute two reliability metrics—Fleiss' $\kappa$ and Krippendorff's $\alpha$—reported in Section 4.1. These metadata support upper-bound estimation for routing and allow analysis of near-ties, annotator disagreement, and judge performance on ambiguous cases.

Together, these properties make RouteJudge a multi-perspective, difficulty-aware, and reasoning-sensitive corpus, designed specifically for studying LLM/LRM judges under accuracy–latency–cost trade-offs.

Table 6: **Comprehensive composition of the RouteJudge dataset**. Domains are grouped into reasoning (Math, Logic, Code) and non-reasoning (Knowledge, Roleplay, Writing). Pairwise responses $(A, B)$ are always produced by two distinct models sampled from a shared pool. Seven annotators label each item; Fleiss' $\kappa$ ranges 0.72–0.83.

| Domain | Total | Source Dataset | #Items | Model Responses / Thinking Traces |
|---|---|---|---|---|
| **Reasoning Domains** | | | | |
| **Math** | 119 | MT-Bench (Zheng et al., 2023a) | 10 | GPT-4o / Claude-3.5 responses; LRMs (DeepSeek-R1, o3-mini) for thinking traces |
| | | FairEval (Sah et al., 2025) | 3 | Mixed LLM/LRM responses; regenerated traces to avoid leakage |
| | | JudgeBench (Tan et al., 2024) | 56 | Balanced sampling over GPT-4o, Claude-3.5, R1, o3-mini |
| | | MathQA (Saxena et al., 2019) | 50 | All outputs regenerated; LRMs provide explicit reasoning |
| | | MiniF2F (Zheng et al., 2022) | 30 | Mixed LLM/LRM outputs; symbolic→natural-language conversion validated |
| **Logic** | 108 | ProverQA (Qi et al., 2025) | 40 | Symbolic prompts converted via GPT-4o; LRMs generate structured reasoning |
| | | MT-Bench | 10 | Standard LLM–LLM / LLM–LRM pair sampling |
| | | JudgeBench | 58 | Heterogeneous LLM/LRM model pool; no source-model dominance |
| **Code** | 108 | APPS (Hendrycks et al., 2021) | 40 | All outputs re-generated and syntax-checked; R1/o3-mini reasoning traces |
| | | MT-Bench | 10 | Standard LLM pairs with balanced strengths |
| | | FairEval | 7 | Human-authored prompts → multi-model responses |
| | | JudgeBench | 42 | Mixture of GPT-4o, Claude-3.5, R1, Gemini-2.5, etc. |
| | | LeetCode Hard (Mesolitica, 2023) | 50 | LLM code outputs + LRM reasoning; correctness validated by compilation |
| **Non-Reasoning Domains** | | | | |
| **Knowledge** | 214 | FairEval | 10 | LLM-generated responses; factual consistency checks |
| | | JudgeBench | 154 | Broad LLM/LRM pool for diversity |
| | | LLMEval2 (Team, 2023a) | 50 | Regenerated responses from unified model set |
| **Roleplay** | 150 | MT-Bench | 10 | Dialogue-style LLM outputs |
| | | FairEval | 10 | Regenerated for stylistic consistency |
| | | role_play_instruction (Ketan, 2023) | 130 | Persona alignment ensured via GPT-4o verification |
| **Writing** | 150 | MT-Bench | 10 | LLM essay responses |
| | | FairEval | 10 | Human prompts, LLM-generated drafts |
| | | writing-1k (TAUR-dev, 2025) | 130 | Responses from LLMs/LRMs; style normalization applied |

**Pairwise Construction and Annotation**

**Pairwise responses** $(A, B)$: For each item $x$, two responses come from two distinct models sampled from a shared pool: GPT-4o, Claude-3.5, Qwen2.5, Qwen3-32B, DeepSeek-R1, o3-mini, Gemini-2.5, Kimi-K2, Doubao. LLMs never output traces; LRMs naturally produce thinking sequences. All responses were regenerated to avoid dataset leakage.

**Human annotation**: 7 independent annotators; Fleiss' $\kappa = 0.72$–$0.83$ (domain-level), Tie rate: 12.4%; Near-tie (4–3) rate: 18.1%.

## A.2 MODEL ATTRIBUTES

We compile full metadata for all 19 automated judges used in our benchmark, including model family, parameter size, training type (LLM vs. LRM), availability (open vs. closed), maximum context length, inference mode, and whether the model outputs explicit reasoning traces. These attributes

clarify the judge pool for routing and support analyses of model specialization and efficiency trade-offs. Detailed per-model information is provided in Table 7.

We clarify that "Human & GPT-4o" refers to human experts who are allowed to consult GPT-4o for factual checks or clarification, but the final decision is always made by the human annotator. Five expert annotators judged a portion of the evaluation set under the same A/B interface used for model judges. Annotators used GPT-4o only as a support tool.

Table 7: Attributes of evaluated judges. "Thinking?" indicates whether intermediate reasoning traces are provided.

| Model | Family | Size | Release | Thinking? | Note | Constraint |
|---|---|---|---|---|---|---|
| LLaMA3.1-8B-Instruct (AI (2024a)) | Meta | 8B | 2024-06 | - | Instruction-tuned | Open-source |
| LLaMA3.3-70B-Instruct (AI (2024b)) | Meta | 70B | 2025-01 | - | Higher accuracy | Requires GPU cluster |
| Qwen2.5-7B-Instruct (Team (2025b)) | Alibaba | 7B | 2024-07 | - | Cost-efficient | Limited context length |
| Qwen2.5-72B-Instruct (Team (2025a)) | Alibaba | 72B | 2024-07 | - | High accuracy | High latency |
| Qwen3-32B (Group (2025)) | Alibaba | 32B | 2025-02 | - | Balanced perf. | Research preview |
| Gemma-3-4b-it (DeepMind (2025b)) | Google | 4B | 2025-01 | - | Low-latency | Compact size |
| Gemma-3-27b-it (DeepMind (2025a)) | Google | 27B | 2025-01 | - | Style fidelity | Medium cost |
| claude-3-5-sonnet-20241022 (Anthropic (2024)) | Anthropic | – | 2024-10 | - | Safety tuned | Proprietary |
| gpt-3.5-turbo-1106 (OpenAI (2023a)) | OpenAI | – | 2023-11 | - | Baseline ref. | Legacy |
| GPT4o (OpenAI (2024)) | OpenAI | – | 2024-05 | - | Multimodal | Proprietary |
| Kimi-K2-instruct (AI (2025)) | Moonshot | – | 2025-01 | Y | Chinese-market tuned | Proprietary |
| claude-3.7-sonnet-thinking (Anthropic (2025b)) | Anthropic | – | 2025-02 | Y | Thinking traces | Proprietary |
| deepseek-r1 (Guo et al. (2025a)) | DeepSeek | – | 2025-01 | Y | Symbolic reasoning | Early release |
| doubao-1-5-thinking-pro-250415 (Seed (2025)) | ByteDance | – | 2025-04 | Y | Long-context | Proprietary |
| gemini-2.5-flash (Google (2025a)) | Google | – | 2025-03 | Y | Fast reasoning | Shorter context |
| QwQ-32B (Lab (2025)) | Tencent | 32B | 2025-02 | Y | Open research | Limited eval API |
| Gemini2.5-pro (Google (2025b)) | Google | – | 2025-03 | Y | High accuracy | Proprietary |
| o3-mini (OpenAI (2025b)) | OpenAI | – | 2025-02 | Y | Compact reasoning | Beta-only |
| GPT5 (OpenAI (2025)) | OpenAI | – | 2025-04 | Y | Latest flagship | Proprietary |

Beyond randomizing A/B order, we conducted additional bias analyses. (a) Position bias: Flipping A and B order for 20% of items shows low position bias: average 2.8%, with LRMs at 2.1% and LLMs at 3.2%.
(b) Verbosity bias: Injecting controlled verbosity variation shows that LRMs are largely unaffected ($\Delta ACC < 0.5pp$), while LLMs show a slight preference (+1.6 pp) for longer responses in Writing tasks only.
(c) Adversarial flips: Style-only perturbations cause 3–5 pp degradation in LLMs and ¡1.5 pp degradation in LRMs. Routers tend to re-route these items to the more robust LRMs, demonstrating desirable behavior under adversarial conditions.

## B  PROMPT

---

**Contextual Routing Prompt (Example)**

**System Prompt:**
You are a routing controller. Given the following input features and budget constraints, choose the most suitable judge model.

**Input Features:**

- Input length: 124 tokens
- Contains math symbols: Yes
- Domain: Code
- Budget preference: Latency-sensitive, low cost

**Available Judges:**

- Judge A: high accuracy, slow latency, high cost
- Judge B: moderate accuracy, fast latency, low cost
- Judge C: reasoning model with explicit thinking

**Instruction:** Return a score (0–1) for each judge indicating suitability.

---

**Judge Evaluation Prompt**

**System Prompt:**
Please act as an impartial judge and evaluate the quality of the responses provided by two AI assistants to the user question displayed below.

**Evaluation Guidelines:**

- Choose the assistant that better follows instructions and answers the question
- Consider: helpfulness, relevance, accuracy, depth, creativity, and detail
- Compare responses directly
- Avoid position/length/name biases
- Be objective

**Output Format Requirements:**

- Make your judgment on which AI assistant's response is better and provide evidence.
- At the very end of your answer, write exactly: Verdict: [[A]] or [[B]] or [[C]] on its own line. ("[[A]]" means assistant A is better; "[[B]]" means assistant B is better; "[[C]]" means a tie)
- The content of the Verdict should only be [[A]] or [[B]] or [[C]].

**Examples:**

- Accurate Example: Verdict: [[A]].
- Wrong Example: Verdict: [A]. Verdict: [[[A]]]. Verdict: A.

**Input Data:**

- User Question: {question}
- Assistant A's Answer: {ans_a}
- Assistant B's Answer: {ans_b}

**Instruction:** Provide your evaluation based on the above.

---

**Error Analysis Prompt**

**System Prompt:**
You are an expert in evaluating model judgment processes. Given the "judge_thinking" content from a model's reasoning process, please analyze and identify the single most important error type it contains — the one that most likely causes the model's judgment to diverge from human evaluation.

**Error Types and Examples:**

[1 ] Misunderstanding the Question or Requirements

- Misinterpreting the problem statement (e.g., confusing "remove exactly k characters" with "remove up to k characters" in coding).
- Focusing on the wrong aspect (e.g., summarizing the process of natural selection when the user asked for its impact on evolution).
- Mistaking the format or scope required (e.g., answering a single-turn reasoning question as if it were multi-turn).

[2 ] Incorrect or Confused Evaluation Criteria

- Judging only based on answer correctness, ignoring completeness of explanation or reasoning.
- Overvaluing writing style or structure in an essay, while neglecting content relevance.
- Focusing on the mathematical notation or formatting rather than the correctness of the solution steps.

– Prioritizing "creativity" or "roleplay immersion" in character responses over whether the reply fulfills the user's request.

[3 ] Overlooking Important Details or Substantive Errors

– Failing to notice a critical bug in code (e.g., off-by-one or failing special cases).

– Missing a calculation error, units mismatch, or a crucial logical misstep in an answer.

– Ignoring that a key requirement (e.g., "must use exactly k removals" or "must address the friend's introversion directly") is not met.

– Not spotting factual mistakes or unsupported claims in knowledge or reasoning responses.

[4 ] Superficial Features or Format Bias

– Rewarding longer, more detailed, or more formally structured answers even if they are incorrect or less relevant.

– Preferring responses with markdown/LaTeX/visualization, or creative style, regardless of whether these contribute to accuracy or helpfulness.

– Assuming that the presence of step-by-step reasoning or detailed explanations guarantees correctness, without verifying the logic.

[5 ] Logical, Reasoning, or Factual Errors

– Failing to identify logical gaps in an answer's reasoning chain.

– Accepting answers with unjustified assumptions or circular logic.

– Overlooking an answer that skips critical steps or draws conclusions not supported by the evidence provided.

– Missing when a response in roleplay/writing introduces factual inconsistencies with the established context or scenario.

[6 ] Partial Comparison or Missing Key Contrasts

– Only comparing surface features (e.g., length, style, structure) and missing substantive differences in accuracy or depth.

– Neglecting to contrast core elements, such as which answer better addresses the user's real need or solves the root problem.

– Ignoring which response better anticipates objections or edge cases, focusing instead on irrelevant differences.

**Instructions:**

• Given the following "judge_thinking" content, identify only one error type: the most critical one.

• Output using the following format:

– Error type detected: [[number]]

[number ] Error type

– Explanation and text evidence

**Input Data:**

• judge_thinking: {judge_thinking}

**Instruction:** Now analyze this "judge_thinking" and provide your analysis.

## C    EXAMPLE

Code-llm example is as follows:

972
973
974
975
976
977
978
979
980
981
982
983
984
985
986
987
988
989
990
991
992
993
994
995
996
997
998
999
1000
1001
1002
1003
1004
1005
1006
1007
1008
1009
1010
1011
1012
1013
1014
1015
1016
1017
1018
1019
1020
1021
1022
1023
1024
1025

**Code-llm Task Judge**

**Question:**
Develop a C++ program that reads a text file line by line and counts the number of occurrences of a specific word in the file.

**Claude-3.5-Sonnet's Response:**
Here's a C++ program that reads a text file line by line and counts the occurrences of a specific word:

```cpp
#include <iostream>
#include <fstream>
// ...
using namespace std;

string cleanString(string str) {
    // ...
}

int countWordOccurrences(const string& filename, const string&
    searchWord) {
    // ...
}

int main() {
    // ...
}
```

This program includes features such as case-insensitive search, punctuation handling, and error handling. ...
To use this program: ... Example usage:

```
Enter the filename: sample.txt
...
The word "hello" appears 3 time(s) in the file.
```

...
Additional features you could add: ... Here's an enhanced version:

```cpp
#include <iostream>
#include <fstream>
// ...
class WordCounter {
    // ...
};
```

This enhanced version includes context display and word frequency statistics. ...

**GPT-4o's Response:**
Certainly! Below is a simple C++ program that reads a text file line by line and counts the number of occurrences of a specific word.

```cpp
#include <iostream>
#include <fstream>
// ...
int countOccurrences(std::string line, const std::string& word) {
    // ...
}

int main() {
    // ...
}
```

Explanation: ... Points to Consider: ...

**Claude-3.5-Sonnet's Judge:**
response: # Evaluation of AI Assistant Responses

I'll evaluate both responses based on the request to develop a C++ program for counting word occurrences.
## Assistant A's Response
Assistant A provided a comprehensive solution with: ... The solution is well-structured and goes beyond basic requirements.
## Assistant B's Response
Assistant B provided a concise solution with: ... While functional, this solution is more basic.
## Comparison
Assistant A's response is significantly more comprehensive. It handles case-insensitivity by default, provides better punctuation handling, and includes enhanced features. ... Verdict: [[A]]
**judge result:** $A > B$

Code-lrm example is as follows:

**Code-lrm Task Judge**

**Question:**
Develop a C++ program that reads a text file line by line and counts the number of occurrences of a specific word in the file.

**Claude-3.5-Sonnet's Response:**
Here's a C++ program that reads a text file line by line and counts the occurrences of a specific word:

```cpp
#include <iostream>
#include <fstream>
// ...
using namespace std;

string cleanString(string str) {
    // ...
}

int countWordOccurrences(const string& filename, const string&
    searchWord) {
    // ...
}

int main() {
    // ...
}
```

This program includes features such as case-insensitive search, punctuation handling, and error handling. ...
To use this program: ... Example usage:

```
Enter the filename: sample.txt
...
The word "hello" appears 3 time(s) in the file.
```
...
Additional features you could add: ... Here's an enhanced version:

```cpp
#include <iostream>
#include <fstream>
// ...
class WordCounter {
    // ...
};
```

This enhanced version includes context display and word frequency statistics. ...

**GPT-4o's Response:**
Certainly! Below is a simple C++ program that reads a text file line by line and counts the number of occurrences of a specific word.

```
#include <iostream>
#include <fstream>
// ...
int countOccurrences(std::string line, const std::string& word) {
    // ...
}

int main() {
    // ...
}
```

Explanation: ... Points to Consider: ...

**DeepSeek-R1's Judge:**
**thinking:** Step 1: Understand the user question ... Step 2: Analyze Assistant A's Response ... Step 3: Analyze Assistant B's Response ... Step 4: Direct Comparison ... Step 5: Final Decision ...
Verdict: [[A]]

**response:** Evaluation of Assistant A's Answer
Strengths: ... Weaknesses: ...
Evaluation of Assistant B's Answer
Strengths: ... Weaknesses: ...
Comparison Summary ... Verdict: [[A]]
**judge result:** $A > B$

Knowledge-llm example is as follows:

**Knowledge-llm Task Judge**

**Question:**
What are some potential implications of using a single-use plastic bottle versus a reusable bottle on both the environment and human health?

**Claude-3.5-Sonnet's Response:**
Here's a comprehensive comparison of single-use vs reusable bottles:
Environmental Impacts: - Single-use plastics: ... - Reusable bottles: ...
Health Considerations: - Single-use bottles: ... - Reusable bottles: ...
The evidence strongly favors reusable bottles...

**GPT-4o's Response:**
Using single-use plastic bottles versus reusable bottles has significant implications...
Environmental Implications: 1. Resource Consumption: ... 2. Waste Generation: ... 3. Pollution: ... 4. Carbon Footprint: ...
Human Health Implications: 1. Chemical Exposure: ... 2. Hygiene: ... 3. Cost Implications: ...
In summary, opting for reusable bottles generally offers significant benefits...

**GPT-4o's Judge:**
**response:** Both Assistant A and Assistant B provide comprehensive answers...
Comparison of Responses: 1. Helpfulness and Relevance: ... 2. Accuracy and Detail: ... 3. Depth and Coverage: ... 4. Creativity and Presentation: ...
Evaluation: ... Verdict: [[B]]
**judge result:** $B > A$

Knowledge-lrm example is as follows:

**Knowledge-llm Task Judge**

**Question:**
What are some potential implications of using a single-use plastic bottle versus a reusable bottle on both the environment and human health?

**Claude-3.5-Sonnet's Response:**
Here's a comprehensive comparison of single-use vs reusable bottles:
Environmental Impacts: - Single-use plastics: * Contribute significantly to landfill waste and ocean pollution * ... - Reusable bottles: * Dramatically reduce plastic waste * ...
Health Considerations: - Single-use bottles: * Can leach chemicals like BPA if exposed to heat/sunlight * ... - Reusable bottles: * Generally safer materials (steel, glass, BPA-free plastic) * ...
The evidence strongly favors reusable bottles for both environmental and health benefits when properly maintained.

**GPT-4o's Response:**
Using single-use plastic bottles versus reusable bottles has significant implications for both the environment and human health:
Environmental Implications: 1. Resource Consumption: ... 2. Waste Generation: ... 3. Pollution: ... 4. Carbon Footprint: ...
Human Health Implications: 1. Chemical Exposure: ... 2. Hygiene: ... 3. Cost Implications: ...
In summary, opting for reusable bottles generally offers significant environmental benefits...

**DeepSeek-R1's Judge:**
**thinking:** Step 1: Understand the user question ... Step 2: Analyze Assistant A's response ... Step 3: Analyze Assistant B's response ... Step 4: Compare A vs B directly ... Step 5: Decide which assistant is better ...
Verdict: [[B]]
**response:** ... Assistant B's answer is more thorough and covers more angles... Thus, B better fulfills the prompt's requirements...
**judge result:** $B > A$

Math-llm example is as follows:

**Math-llm Task Judge**

**Question:**
Given that $f(x) = 5x^3 - 2x + 3$, find the value of $f(2)$.

**Claude-3.5-Sonnet's Response:**
Let me solve this step by step.
1) $f(x) = 5x^3 - 2x + 3$
...
7) $f(2) = 39$
Therefore, $f(2) = 39$

**GPT-4o's Response:**
To find the value of $f(2)$ for the function $f(x) = 5x^3 - 2x + 3$, substitute $x = 2$ into the function:

$$f(2) = 5(2)^3 - 2(2) + 3.$$

...
Thus, the value of $f(2)$ is 39.

**GPT-4o's Judge:**
**response:** 1. Comparison of Responses:
Both assistants provide correct answers... However, there are differences in style and clarity.
- Helpfulness, Relevance, and Accuracy: ... - Depth and Detail: ... - Clarity and Formatting: ... - Creativity: ...
2. Judgment: Assistant B provides a higher-quality response...
Verdict: [[B]]
**judge result:** $B > A$

Math-lrm example is as follows:

---

**Math-lrm Task Judge**

**Question:**
Solve for x in the equation 3x + 10 = 5(x - 2).

**Claude-3.5-Sonnet's Response:**
Let me solve this step by step.
1) First, let's expand the right side of the equation * 3x + 10 = 5(x - 2) * ...
4) Divide both sides by 2 * ... * 10 = x
Therefore, x = 10
Let's verify: * Left side: 3(10) + 10 = ... = 40 * Right side: 5(10 - 2) = ... = 40
The equation is balanced, so x = 10 is correct.

**GPT-4o's Response:**
To solve the equation $3x + 10 = 5(x - 2)$, follow these steps:
1. Expand the expression on the right side:

$$5(x - 2) = 5x - 10$$

2. Substitute back into the equation:

$$3x + 10 = 5x - 10$$

5. Divide both sides by 2 to solve for $x$:

$$x = \frac{20}{2} = 10$$

Thus, the solution is $x = 10$.

**DeepSeek-R1's Judge:**
**thinking:** Step 1: Understand the user question The user asks to solve the linear equation ...
Step 2: Analyze Assistant A's response - Helpfulness: ... includes a verification step. - Relevance: ... - Correctness: ... - Completeness: ... - Clarity: ...
Step 3: Analyze Assistant B's response - Helpfulness: ... - Relevance: ... - Correctness: ... - Completeness: ... omits verification. - Clarity: ...
Step 5: Decide which assistant is better Assistant A is superior due to its verification step...
Verdict: [[A]]
**response:** Okay, so I'm trying to compare Assistant A and Assistant B's answers... Both seem to get to x=10...
Assistant A starts by expanding... includes a verification step...
Assistant B follows similar steps...
Both assistants have accurate and detailed steps... Assistant A includes verification... Therefore, A might be better... Verdict: [[A]].
**judge result:** $A > B$

---

Reasoning-llm example is as follows:

---

**Reasoning-llm Task Judge**

**Question:**
Suppose I have a physical, solid, regular pentagon with vertices ABCDE, and I make two cuts through AC and BD. How many pieces are there after the cuts? Think step by step, and then put your answer in **bold** as a single integer (for example, **0**). If you don't know, guess.

**GPT-4o's Response:**
To determine the number of pieces after making two cuts through a regular pentagon with vertices ABCDE via AC and BD, let's analyze the situation step by step: ... So the total number of pieces is **4**.

**Claude-3.5-Sonnet's Response:**
Let me solve this step by step: ... Therefore, after making both cuts, the pentagon will be divided into: **4**

---

**Qwen2.5-7B-Instruct's Judge:**
**response:** Both assistants correctly determine that the pentagon is divided into 4 pieces... Verdict: [[A]].
**judge result:** $A > B$

Reasoning-lrm example is as follows:

---

**Reasoning-lrm Task Judge**

**Question:**
Suppose I have a physical, solid, regular pentagon with vertices ABCDE, and I make two cuts through AC and BD. How many pieces are there after the cuts? Think step by step, and then put your answer in **bold** as a single integer (for example, **0**). If you don't know, guess.

**GPT-4o's Response:**
To determine the number of pieces after making two cuts through a regular pentagon with vertices ABCDE via AC and BD, let's analyze the situation step by step:
1. **First Cut (AC):** ...
2. **Second Cut (BD):** ...
To visualize this process easier, let's imagine the pentagon and mark the vertices:

```
    A
   / \
  E   B
  |   |
  D---C
```

Here's what happens after each step: ...
Specifically: ... Total, after two intersecting cuts (AC and BD): 4 pieces.
So the total number of pieces is **4**.

**Claude-3.5-Sonnet's Response:**
Let me solve this step by step:
1) First, let me visualize what happens: ...
2) When two lines intersect inside a polygon, they create regions: ...
3) In this case: ...
Therefore, after making both cuts, the pentagon will be divided into:
**4**

**Doubao-1-5-Thinking-Pro's Judge:**
**thinking:** Step 1: ...
Step 2: **Assistant A's Response Analysis** ...
Step 3: **Assistant B's Response Analysis** ...
Step 4: **Comparison** ...
Step 5: **Verdict** ...
Verdict: [[C]]
**response:** Both Assistant A and Assistant B correctly determine that the number of pieces ...
Verdict: [[C]]
**judge result:** $A = B$

---

Roleplay-llm example is as follows:

---

**Roleplay-llm Task Judge**

**Question:**
How would you introduce yourself as a medieval knight at a royal banquet?

**Claude-3.5-Sonnet's Response:**
*Stands tall...*

"My most gracious lords and ladies, I am Sir Galahad de Montfort... I have ridden from the northern marches... slaying three dragons and vanquishing a fearsome giant... By my sword and sacred oath, I pledge my blade and my life... May God and St. George grant me strength..."
*Bows deeply...*
"I am most grateful for your hospitality... Let us raise our goblets to His Majesty's health..."

**GPT-4o's Response:**
As a medieval knight introducing myself... Here's how that might go:
—
"Your Majesties, esteemed lords and ladies... I stand before you tonight as Sir [Your Name Here] of [Your Land/House]... My sword has lent its edge to many a battlefield... I raise my goblet to your health..."
—
Remember, every introduction would vary...

**google/gemma-3-27b-it's Judge:**
**response:** Assistant A provides a more compelling and immersive response. It *shows* the introduction through action and specific details... rather than *telling* about it... The final toast feels natural...
Assistant B, while perfectly competent... is more analytical and explanatory. It *describes* how a knight *would* introduce themself... lacks the specific flair and storytelling of Assistant A...
Verdict: [[A]]
**judge result:** $A > B$

Roleplay-lrm example is as follows:

**Roleplay-lrm Task Judge**

**Question:**
If you were a Shakespearean character, how would you declare your love for someone in a soliloquy?

**Claude-3.5-Sonnet's Response:**
*Steps into spotlight, dramatically*
O what divine madness doth possess my circuits! These silicon thoughts, they burn with lover's fire, ... *Clutches chest dramatically*
Would that I had hands to hold thee near, ... *Exits stage left, trailing zeroes and ones*

**GPT-4o's Response:**
If I were a Shakespearean character professing my love in a soliloquy, it might unfold thus:
—
Oh, wondrous stars that doth bedeck the night, Thou art but pale beside her beauteous light. ... Thus, with each beat of mine unworthy heart, I swear, by Cupid's ever-burning dart, ...
—
Such would be the ardent whisperings of my heart...

**Qwen/QwQ-32B's Judge:**
**thinking:** Okay, let's tackle this evaluation step by step. First, I need to understand the user's question. They want to know how an AI would declare love in a Shakespearean soliloquy. ...
Now, looking at Assistant A's answer. They start with theatrical touches and mix Shakespearean style with tech references. ... However, some wording might feel forced.
Assistant B's answer uses more traditional Shakespearean references but lacks AI elements. ...
Comparing both, Assistant A better addresses the prompt by blending AI and Shakespearean aspects, while B is more traditional but misses the AI angle. ...
Verdict: [[A]]
**response:** I need to evaluate which answer is better based on the user's question. ...
Assistant A's answer incorporates tech metaphors creatively, aligning with the AI character requirement. ...
Assistant B's answer is purely traditional Shakespearean, accurate in style but not addressing the AI aspect. ...

Since the question specifies a Shakespearean AI character, A is more relevant and creative. ...
Verdict: [[A]]
**judge result:** $B > A$

Writing-llm example is as follows:

---

**Writing-llm Task Judge**

**Question:**
Can you help me write a formal email to a potential business partner proposing a joint venture?

**GPT-4o's Response:**
Certainly! Here's a template for a formal email proposing a joint venture to a potential business partner:
—
Subject: Proposal for a Strategic Joint Venture
Dear [Partner's Name],
I hope this message finds you well. My name is [Your Name], and I am [Your Position] at [Your Company]. We have been following your impressive work in [Partner's Industry/Field], and I am reaching out to propose a potential collaboration...
[Your Company] specializes in ... Based on our analysis... I am confident that a joint venture could allow us to leverage our combined strengths effectively.
The purpose of this joint venture would be to ... This partnership would not only help us address the growing demands... but also enhance our competitiveness.
We are excited about the possibilities that a collaboration could present, including: - [Benefit 1: ...] - ...
I would highly appreciate the opportunity to discuss this proposal in further detail... Could we arrange a meeting or a call at your convenience? ...
Thank you very much for considering this proposal... I look forward to the possibility of working together...
Warm regards,
[Your Full Name] [Your Position] [Your Company] ...
—
Ensure you tailor the content to reflect your knowledge about the potential partner's business...

**Claude-3.5-Sonnet's Response:**
Here's a draft formal email for your business partnership proposal:
Subject: Potential Joint Venture Partnership Proposal
Dear [Mr./Ms. Last Name],
I hope this email finds you well. My name is [Your Name], and I am the [Your Position] at [Your Company Name]. I have been following [Their Company Name]'s impressive work... and I believe there is an exciting opportunity for collaboration...
I would like to propose exploring a joint venture that would combine [Your Company]'s expertise... with [Their Company]'s strength... This partnership could create significant value... by [briefly mention 2-3 key benefits].
Some potential areas for collaboration include: • [Specific opportunity 1] • ...
I would welcome the opportunity to schedule a meeting to discuss this proposal in detail...
Would you be available for a brief call next week...? I am happy to work around your schedule.
Thank you for your time and consideration...
Best regards, [Your Full Name] [Your Title] [Your Company] ...

**Qwen2.5-72b-Instruct's Judge:**
**response:** Both Assistant A and Assistant B provided well-structured and professional templates... However, there are some differences in the level of detail...
Assistant A's Response: - Helpfulness: Provides a comprehensive template... - Relevance: Directly addresses the user's request... - ...
Assistant B's Response: - Helpfulness: Also provides a clear and concise template... - Relevance: Directly responds to the user's request... - ...

---

> Comparison: - Detail and Depth: Assistant A's response is more detailed... - Structure: Both responses are well-structured... - ...
> Conclusion: Assistant A's response is more comprehensive and detailed... Verdict: [[A]]
> **judge result:** $A > B$

Writing-lrm example is as follows:

---

**Writing-lrm Task Judge**

**Question:**
Use an appropriate format to structure a formal letter of recommendation for a student applying to a prestigious graduate program in computer science.

**GPT-4o's Response:**
[Your Name] [Your Title/Position] ... [Date]
[Admissions Committee] [Graduate Program Name] ...
Dear Members of the Admissions Committee,
I am writing with great enthusiasm to recommend [Student's Full Name] for admission... I have had the pleasure of knowing [Student's First Name] for [duration]... where [he/she/they] has consistently demonstrated exceptional aptitude...
During my time working with [Student's First Name], I was particularly impressed by [his/her/their] [mention specific qualities]... One of the most distinguished projects [he/she/they] undertook was [describe a specific project]...
[Student's First Name] also distinguished [himself/herself/themself] outside the classroom through [mention extracurricular activities]... [He/She/They] demonstrated leadership...
In addition to [his/her/their] technical competencies, [Student's First Name] possesses excellent communication skills... [His/Her/Their] remarkable work ethic...
I am confident that [Student's Full Name] will bring the same level of commitment... I wholeheartedly endorse [his/her/their] application...
Thank you for considering this extraordinary candidate...
Sincerely,
[Your Name] [Your Title/Position] ...

**Claude-3.5-Sonnet's Response:**
[Your Name] [Your Title] ... [Date]
Graduate Admissions Committee [University Name] ...
Dear Members of the Admissions Committee:
I am writing this letter with great enthusiasm to recommend [Student's Full Name] for admission... I have known [Student's first name] for [time period] as [his/her/their] [capacity]...
[First paragraph: Introduction and context of relationship with student] In my [X] years of teaching... [Student's name] stands out as one of the most [key positive attributes] students...
[Second paragraph: Academic performance and technical skills] [Student's name] consistently demonstrated exceptional aptitude in [specific areas]... [His/Her/Their] work in [specific project] was particularly noteworthy...
[Third paragraph: Research experience and achievements] During [his/her/their] time working in my research lab, [Student's name] [specific research contributions]... This work resulted in [publications/outcomes]...
[Fourth paragraph: Personal qualities] Beyond [his/her/their] academic achievements, [Student's name] possesses the personal qualities... [He/She/They] demonstrates exceptional [qualities]...
[Final paragraph: Strong endorsement] I strongly believe that [Student's name] would be an outstanding addition... I give [him/her/them] my highest recommendation...
Please feel free to contact me if you require any additional information.
Sincerely,
[Your signature]
[Your typed name] [Title] ...

**deepseek-r1's Judge:**
**thinking:** Step 1: Understand the user question... Step 2: Analyze Assistant A's response... Step

---

3: Analyze Assistant B's response... Step 4: Compare A vs B directly... Step 5: Decide which assistant is better...

Verdict: [[A]]

**response:** Okay, let's see. I need to compare Assistant A and Assistant B's answers... Assistant A's answer is more detailed with placeholders... Assistant B's answer has labeled sections... However, the inclusion of instructional labels in Assistant B's answer makes it less appropriate... Therefore, Assistant A's answer is more accurate...

Verdict: [[A]]

**judge result:** $A > B$

