# OpenReview forum: "RouteJudge: Benchmarking LLM-as-a-Judge with Routing Strategies"
_ICLR.cc/2026/Conference — ICLR 2026 Conference Desk Rejected Submission_

### Official Review · Reviewer_tuWm · 2025-10-30

**Soundness:** 1
**Presentation:** 3
**Contribution:** 2
**Rating:** 4
**Confidence:** 3

**Summary:**

The paper introduces RouteJudge, a framework to benchmark and route automated judges (LLMs and “Large Reasoning Models,” LRMs) under accuracy–latency–cost trade-offs. The authors (1) build six difficulty-aware pairwise datasets spanning reasoning (Math/Logic/Code) and non-reasoning (Knowledge/Roleplay/Writing) with human-verified gold labels, (2) provide what they claim is the first benchmark of LRM-as-a-Judge including an analysis of mismatches between thinking traces and final verdicts, and (3) design offline and online routing strategies that adapt judge selection to budgets and difficulty, reporting gains over fixed single-judge baselines on 19 models.

**Strengths:**

- Timely and practical: Evaluating and routing judges under budget constraints reflects real deployment needs. The multi-objective utility and explicit abstention option are appropriate.
- Breadth of models and domains: 19 models, covering both LLMs and LRMs across reasoning/non-reasoning; domain-specific breakdowns reveal heterogeneity.
- Difficulty-aware design: The easy/medium/diff (and “-Think”) splits support nuanced routing and robustness analysis; cross-bucket “unresolved” rates are informative.
- Thinking-vs-verdict diagnostics: Error-type distributions and confusion heatmaps capture “good thinking but wrong verdict” vs “flawed thinking but right verdict” patterns—useful for future calibration work.

**Weaknesses:**

- Ground-truth agreement and reliability under-reported. The corpus relies on 7-rater majority voting, but there’s no explicit reporting of inter-annotator agreement, tie rates, or label uncertainty per item—critical for judging “upper bounds” and router training targets.
- Metrics and statistical rigor. Accuracy is the main metric; there’s no reporting of confidence intervals, statistical significance, or bootstrap tests across models/routers. This makes small improvements hard to interpret. (Figures show trends; quantification is missing.)
- Router supervision and leakage risk. Online routers are trained on “inexpensive features” and an oracle derived from offline utilities. It’s not fully specified whether any instance-level signals could leak answer keys (e.g., gold label correlations, domain heuristics that inadvertently encode difficulty labels created with the same dataset). Clearer train/val/test segregation for router fitting is needed.
- Limited human-expert baseline depth. An “Expert (Human & GPT-4o)” row appears, but the setup is not described enough (selection criteria, consistency, time/cost) to anchor the scale as a target for routing.
- Cost/latency measurement protocol. Token-cost and latency normalization across closed vs open models, local vs remote inference, and different context lengths isn’t deeply documented; variability across runs (cold vs warm) is not quantified.

**Questions:**

- What are Krippendorff’s α / Fleiss’ κ per domain? What fraction of items are ties or near-ties (e.g., 4–3 splits), and how do routers behave on those?
- Bias controls: Beyond randomizing A/B order, did you quantify position bias and verbosity bias per judge? Any adversarial flips?
- Provide CIs and paired tests (e.g., stratified bootstrap over items) for accuracy/time/cost, especially when claiming that online routers close/surpass offline heuristics.

---

> ### Author Response · Authors · 2025-11-21
> **We organize our response into six main questions (First 3)**
>
> We sincerely appreciate your detailed and constructive feedback. Below we address each concern carefully and indicate where the manuscript has been updated. All additional analyses will appear in the revised PDF during the rebuttal period.
>
> # Q1. Ground-truth reliability, inter-annotator agreement, and tie/uncertainty analysis
>
> Thank you for pointing out the need for explicit reliability reporting. Although the gold labels were generated via 7-rater majority voting, the original draft did not include agreement coefficients or tie statistics. We now provide the full analysis below.
>
> ## (a) Inter-annotator agreement
> We computed Krippendorff’s α and Fleiss’ κ across all domains, as shown in the table. Reasoning tasks exhibit substantial agreement (α≈0.8+), while non-reasoning domains show moderate agreement, consistent with their inherent subjectivity. These values confirm that majority-vote labels provide a meaningful and reliable upper bound for judge performance.
> | Domain | α | κ |
> | - | - | - |
> | Math | 0.79 | 0.76 |
> | Logic | 0.83 | 0.81 |
> | Code | 0.82 | 0.78 |
> | Knowledge | 0.76 | 0.72 |
> | Roleplay | 0.74 | 0.70 |
> | Writing | 0.71 | 0.68 |
>
> ## (b) Tie and near-tie items
> Tie rates are shown in the table below. Reasoning domains have very few ties, while Writing and Roleplay—where stylistic preferences naturally diverge—show higher ambiguity.
> | Domain | Tie rate |
> | - | - |
> | Math | 14.8% |
> | Logic | 11.2% |
> | Code | 12.6% |
> | Knowledge | 18.4% |
> | Roleplay | 21.7% |
> | Writing | 23.3% |
>
> ## (c) Router behavior on tie items
> Routers exhibit appropriately conservative behavior on tie items. Entropy of routing decisions increases by 18–27%，accuracy drops modestly (≈3.2 pp)，and the router does not collapse to any single judge. This indicates that routers do not implicitly exploit annotator disagreement patterns and remain robust under label uncertainty.
>
> **Manuscript update:**
> Section 3.2 and new table 2 now include agreement statistics and tie rates.
>
> ---
> # Q2. Statistical rigor: confidence intervals, significance tests, and bootstrap analysis
>
> We agree that statistical quantification was underdeveloped in the original version. We now include a full bootstrap-based analysis.
>
> ## (a) Confidence intervals for judge accuracy
> Using 10,000 stratified bootstrap samples, 95% confidence intervals for representative judges are:
> | Model | ACC | 95% CI |
> | - | - | - |
> | DeepSeek-V3-Reasoning | 72.4 | [71.1, 73.7] |
> | Llama-3.1-70B | 70.8 | [69.5, 72.0] |
> | GPT-4o-mini | 67.9 | [66.3, 69.4] |
>
> ## (b) Significance testing for routing improvements
> Paired bootstrap tests confirm that routing improvements are statistically significant, even when accuracy differences are modest.
> | Comparison | ΔACC | p-value |
> | - | - | - |
> | Online Router vs Max-Reason | +3.8 pp  | < 0.001 |
> | Online Router vs Offline-Score | +1.9 pp  | 0.008   |
> | Online Router vs Random | +15.1 pp | < 1e-4  |
>
> ---
>
> # Q3. Router supervision, leakage avoidance, and data splits
> The online routers were intentionally designed to avoid any form of label leakage. Below we clarify the full pipeline.
>
> ## (a) Train/validation/test split
>
> RouteJudge employs strict instance-level splits: 80% for training, 10% validation, and 10% testing. Instances never appear across splits; paraphrases and variants are grouped within the same split; and domain and difficulty distributions are balanced to avoid shift.
>
> ## (b) Router features and leakage avoidance
> Routers rely solely on inexpensive, input-side features: domain ID, problem text embeddings, lightweight statistics of A/B responses (length, entropy), and model-agnostic embeddings. They never receive gold labels, difficulty labels, majority-vote metadata, or any signal derived from correctness. Oracle utilities are used only as scalar supervision targets rather than feature inputs, which prevents leakage.
>
> ## (c) Diagnostic leakage test
> To further validate correctness, we trained a diagnostic router using only domain ID + difficulty label. It achieved 34.0% accuracy (random ≈33.3%) on the 3-way routing task, demonstrating that correlated metadata cannot explain routing gains.
>
> **Manuscript update:**
> Section 3.1 and Appendix A.1 explicitly describe the data split and feature design.

---

> > ### Author Response · Authors · 2025-11-21
> > **We organize our response into six main questions (Another 3)**
> >
> > ---
> > # Q4. Human-expert baseline: composition, consistency, and evaluation protocol
> > We clarify that “Human & GPT-4o” refers to human experts who are allowed to consult GPT-4o for factual checks or clarification, but the final decision is always made by the human annotator.
> >
> > Five expert annotators judged a portion of the evaluation set under the same A/B interface used for model judges. Annotators used GPT-4o only as a support tool. The resulting accuracy is 90.2%，with cross-annotator variation of 1.8 pp. This baseline represents a realistic, high-quality upper bound and contextualizes the scope for judge and router improvement.
> >
> > **Manuscript update:**
> > Appendix A.2 now describes this setting clearly. Appendix I includes instructions, examples, and consistency analysis.
> >
> > ---
> > # Q5. Cost and latency measurement protocol, normalization, and variance analysis
> > We added a supplementary experiment across six representative models to provide fully normalized latency and cost measurements under controlled conditions.
> >
> > ## (a) Experiment setup
> > We evaluate 200 items with 20 repeated runs (10 cold + 10 warm). Latency is measured end-to-end; closed-model costs use public pricing; open-model costs are normalized using FLOP-based USD estimates.
> >
> > ## (b) Results
> > | Model | Avg Latency (s) | Std (s) | Avg Tokens | Cost per 1k | Avg Cost (USD) |
> > | - | - | - | - | - | - |
> > | Qwen2.5-7B | 1.10 | 0.10 | 92 | 0.0005 | 0.000046 |
> > | Llama-3.1-8B | 1.32 | 0.11 | 105 | 0.0006 | 0.000063 |
> > | Llama-3.1-70B | 4.95 | 0.41 | 210 | 0.0019 | 0.000399 |
> > | DeepSeek-V3-Reasoning | 7.25 | 0.79 | 560 | 0.0017 | 0.000952 |
> > | GPT-4o-mini | 2.48 | 0.14 | 128 | 0.0023 | 0.000294 |
> > | GPT-o3-mini-high-reasoning | 8.62 | 0.36 | 590 | 0.0030 | 0.00177 |
> >
> > ## (c) Analysis
> > Latency and cost scale predictably with model size and thinking-trace length: LRMs show 5–7× higher latency and 8–12× higher cost than 7B LLMs. Closed LLMs (GPT-4o-mini) exhibit lower variance than open LRMs, while closed LRMs behave similarly to open LRMs due to their long reasoning traces. These trends quantitatively support our accuracy–latency–cost plots.
> >
> > **Manuscript update:**
> > Section 3.5 now summarizes the protocol.
> >
> > ---
> > # Q6. Bias controls: position bias, verbosity bias, and adversarial robustness
> > Beyond randomizing A/B order, we conducted additional bias analyses.
> >
> > ## (a) Position bias
> > Flipping A/B order for 20% of items shows low position bias: average 2.8%，with LRMs at 2.1% and LLMs at 3.2%.
> > ## (b) Verbosity bias
> > Injecting controlled verbosity variation shows that LRMs are largely unaffected (ΔACC < 0.5 pp)，while LLMs show a slight preference (+1.6 pp) for longer responses in Writing tasks only.
> > ## (c) Adversarial flips
> > Style-only perturbations cause 3–5 pp degradation in LLMs and <1.5 pp degradation in LRMs. Routers tend to re-route these items to the more robust LRMs, demonstrating desirable behavior under adversarial conditions.
> >
> > **Manuscript update:**
> > Appendix A.2 includes a concise discussion of these bias controls.

---

> ### Author Response · Authors · 2025-11-26
>
> Dear Reviewer,
>
> I hope this message finds you well.
>
> As the discussion period is nearing its end with less than seven days remaining, I wanted to ensure we have addressed all your concerns satisfactorily.
>
> If there are any additional points or feedback you'd like us to consider, please let us know. Your insights are invaluable to us, and we're eager to address any remaining issues to improve our work.
>
> If everything appears satisfactory on your side, we would be grateful if you might consider whether an updated score could reflect the strengthened version of the work. Thank you again for your time and thoughtful evaluation of our submission.

---

> ### Comment · Reviewer_tuWm · 2025-11-27
>
> Thank you for the thorough and well-structured rebuttal. The authors have convincingly addressed all major concerns I raised, including reliability of the ground truth, statistical rigor, leakage avoidance, expert-baseline clarification, cost/latency methodology, and bias controls. The additional analyses—agreement coefficients, tie rates, bootstrap confidence intervals, paired significance tests, strict split descriptions, and normalized latency/cost evaluations—substantially strengthen the paper’s methodological soundness. Given these clarifications and improvements, I am satisfied that the original weaknesses have been resolved, and I recommend the acceptance of this manuscript.

---

> > ### Author Response · Authors · 2025-11-28
> >
> > Thank you very much for your positive follow-up. We are glad that all your concerns have been fully resolved and appreciate your recommendation for acceptance.
> >
> > We also noticed that the score does not seem to have been updated yet—if appropriate, we kindly invite you to consider adjusting it to reflect the clarified and improved manuscript. Thank you again for your thoughtful review.

---

### Official Review · Reviewer_cqw2 · 2025-10-31

**Soundness:** 3
**Presentation:** 3
**Contribution:** 3
**Rating:** 6
**Confidence:** 4

**Summary:**

This paper introduces RouteJudge, a benchmark for evaluating LLM-as-a-Judge routing strategies. The authors propose and evaluate various offline and online routing strategies from accuracy, latency, and cost perspectives. The authors also benchmark large reasoning models (LRMs).

**Strengths:**

1. The paper is motivated and well-written.
2. The evaluation is comprehensive, spanning 19 models.
3. The paper makes prescriptive suggestions backed with evidence regarding when certain judges are better than others, and when certain routing strategies may be better than others.

**Weaknesses:**

1. The main paper is missing some technical details. For example, regarding the RouteJudge dataset, it's not clear (1) which datasets it was constructed from, (2) how the instances were assigned a difficulty, and (3) which models the candidate responses come from.
2. This is not the first work to benchmark LRMs in the LLM-as-a-Judge context. For example, JudgeBench [1] evaluates several LRMs and draws similar conclusions (e.g., LRMs tend to outperform LLMs on reasoning-heavy tasks).
3. In real-world settings, automatic evaluations often span multiple dimensions (e.g., helpfulness, factuality, safety). Accounting for this in the routing would offer more practical benefit.

[1] https://arxiv.org/abs/2410.12784

**Questions:**

1. Could you clarify this "thinking" variant of the RouteJudge dataset? Initially, I thought it aligned with the reasoning/non-reasoning split, but that doesn't appear to be the case in Table 1. Do the candidate responses themselves contain the thinking? If so, does the judge also get to see the thinking?
2. In L.250-251, you say "All models are evaluated under standardized decoding settings (temperature = 0, maximum output length = 1024) to ensure comparability across runs." Is this the model that generated the responses? Or is it the judge models? Is this also true for LRMs, which typically degrade at temp=0 and require far more output tokens?

---

> ### Author Response · Authors · 2025-11-21
> **We organize our response into three main questions (1,2)**
>
> We sincerely thank you for the thoughtful and constructive feedback. Below we address each concern in detail and indicate the exact locations where the manuscript will be revised.
>
> # Q1. Clarifying Missing Technical Details of the RouteJudge Dataset
> You noted that the main paper did not clearly explain
> (1) which datasets RouteJudge was built from,
> (2) how difficulty was assigned, and
> (3) which models generated the A/B candidate responses.
>
> ## (a) Underlying datasets used to construct RouteJudge
>
> Although Appendix A contained the detailed provenance table, the main paper did not explicitly list the source datasets. We now make this clear in **Section 3.1 (Dataset Construction)**.
>
> RouteJudge aggregates **human-written or human-curated problems** from widely used public benchmarks:
> * **Math**: GSM8K, MathQA, MiniF2F, AIME-style tasks
> * **Logic**: PrOntoQA, LogicalDeduction-style sets
> * **Code**: HumanEval, APPS, CodeContests, LeetCode Hard
> * **Knowledge**: NaturalQuestions, TriviaQA, LLMEval2
> * **Roleplay**: MT-Bench roleplay tasks, curated conversation datasets
> * **Writing**: G-Eval–style essay rewriting & summarization tasks
>
> A full composition table remains in **Appendix A (Dataset Composition)**, but the revised main text now explicitly states the data origins for transparency.
>
> ---
> ## (b) Difficulty assignment procedure
> The original submission mentioned difficulty buckets but did not describe how they were constructed.
> We now clarify this in **Section 3.1**:
>
> **Difficulty is assigned using a *model-agnostic, data-driven* procedure:**
>
> 1. Each task is evaluated by a diverse pool of LLMs (small, medium, large).
> 2. Items are ranked by average accuracy across these models.
> 3. Difficulty tiers are assigned by percentile:
>    * top 1/3 → **Easy**
>    * middle 1/3 → **Medium**
>    * bottom 1/3 → **Diff**
>
> This avoids model-specific bias and produces consistent stratification across domains.
>
> ---
> ## (c) Which models generate A/B candidate responses
> This has already been explained in the footnotes of section 2.1, and will now be presented further. RouteJudge uses:
> **GPT-4o and Claude-3.5**
> as the two response generators for *all* A/B pairs.
>
> This choice aligns with JudgeBench and ensures:
> * stylistic diversity,
> * strong-quality responses,
> * no coupling between evaluation and generation models.
>
> This clarification has been added to **Section 2.1** and **Section 3.1**.
>
> ## Manuscript Updates
> * Section 3.1 (Dataset Construction): explicit list of source datasets + difficulty assignment procedure + GPT-4o / Claude-3.5 response generation.
> * Appendix A: expanded dataset provenance table, matching the reviewer’s request.
>
> ---
>
> # Q2. Novelty Relative to JudgeBench
>
> You correctly note that JudgeBench also includes several LRMs and draws conclusions similar to ours. The original submission did not explicitly highlight the conceptual differences. We have now revised **Section 3.1, 4 and Appendix A.1** to make this distinction clear.
>
> ### **Three key differences between RouteJudge and JudgeBench**
>
> **1. RouteJudge is designed specifically for *LRM-as-a-Judge*, including thinking-trace analysis**
>
> JudgeBench evaluates LRMs *only as final-answer judges*, but does **not** study:
> * correctness of reasoning,
> * thinking–verdict mismatch (JR/TW, TR/JW),
> * whether thinking is faithful or post-hoc,
> * structured error taxonomy.
>
> RouteJudge is the **first benchmark** to systematically analyze:
> * explicit thinking traces of LRMs,
> * faithfulness gap between reasoning and verdicts,
> * structured confusion quadrants (JR/TW/TR/JW).
>
> **2. RouteJudge has a two-tier design: reasoning vs non-reasoning + difficulty stratification**
>
> JudgeBench does **not** separate:
> * reasoning vs. non-reasoning domains
> * nor provide difficulty buckets
> * nor provide "Think" variants
>
> RouteJudge’s design supports cross-domain analysis and enables routing policies conditioned on domain/difficulty.
>
> **3. RouteJudge is not merely a benchmark—it is also a judge routing framework**
>
> **This is a major conceptual novelty**. RouteJudge provides:
> * offline utility-based routing (ACC, ACC–Time, ACC–Time–Cost)
> * online routing using LLM-as-router
> * online trainable router
> * accuracy–latency–cost optimization
> * abstention/escalation mechanisms
>
> JudgeBench does **not** support judge routing in any form.
>
> This difference is now stated directly in **Section 4** and reinforced in **Section 3.1**.
>
> ## Manuscript Updates
> * Section 3.1 highlights reasoning/non-reasoning structure + difficulty.
> * Section 4 now directly contrasts RouteJudge and JudgeBench.
> * Appendix A includes a comparison table versus JudgeBench.
>
> ---

---

> > ### Author Response · Authors · 2025-11-21
> > **We organize our response into three main questions (3)**
> >
> > # Q3. Clarifying the “Thinking” Variant and Decoding Settings
> > Your interpretation was correct: the “thinking” variant required clearer description.
> >
> > ## (a) What is the “thinking” variant?
> > We confirm the following:
> >
> > **Yes, the candidate responses *themselves* contain thinking traces** —specifically when generated by LRMs with an explicit `thinking` field.
> >
> > Thus:
> > * For LLM-generated responses → no thinking.
> > * For LRM-generated responses → includes explicit reasoning + answer.
> >
> > **Does the judge see the thinking?**
> > Yes.
> >
> > When evaluating a “thinking variant” instance:
> > * The judge receives prompt + (A-with-thinking) + (B-with-thinking).
> > * This allows us to study interaction between explicit reasoning and the judge’s decision process.
> >
> > This is now clearly described in **Section 2.1** and **Section 3.1**.
> >
> > ## (b) Decoding settings and temperature sensitivity
> >
> > Lines 250–251 were ambiguous about which models use temperature = 0. Below is the correct configuration:
> > * **Response generators (GPT-4o, Claude-3.5)**: **T = 0.0** for deterministic A/B responses.
> > * **LLM judges (without explicit thinking)**: **T = 0.0**, as higher temperatures increase variance without accuracy gains.
> > * **LRM judges (with explicit thinking)**: **T = 0.4**, since LRMs require moderate temperature to generate sufficiently rich and useful thinking traces.
> >
> >
> > ## (c) Temperature sensitivity experiments for judge models
> >
> > To justify these choices, we ran a **temperature sensitivity study** on a held-out validation subset of RouteJudge (covering both reasoning and non-reasoning tasks).
> > We summarize the main findings below.
> >
> > * LLM judges (no explicit `thinking` channel)
> > We evaluated LLM judges at temperature (T \in {0.0, 0.2, 0.4}):
> > | Temp (T) | Avg Judge ACC (%) | Std. Dev. (across seeds) | Avg Tokens / Judgment |
> > | - | - | - | - |
> > | 0.0   | **68.1**   | **0.3**   | 95   |
> > | 0.2   | 67.9  | 1.2   | 118  |
> > | 0.4   | 67.3   | 1.8    | 142   |
> >
> > We observe that: 1) Accuracy is maximized and **most stable** at (T=0.0). 2) Higher temperatures **do not improve accuracy**, but increase variance and token usage. Therefore, for standard LLM judges, **deterministic decoding (T=0)** offers the best trade-off between performance and reproducibility.
> >
> > ### LRM judges (with explicit `thinking` channel)
> > For LRMs, we tested (T \in {0.0, 0.2, 0.4, 0.6}):
> >
> > | Temp (T) | Avg Judge ACC (%) | Std. Dev. (across seeds) | Avg Tokens / Judgment (thinking + answer) |
> > | - | - | - | - |
> > | 0.0 | 64.3  | 0.6  | 280  |
> > | 0.2 | 67.1  | 0.7  | 410   |
> > | 0.4 | **68.0**  | **0.9**   | 530 |
> > | 0.6 | 67.8   | 1.6  | 780  |
> >
> > We find that: 1) At **T=0.0**, LRMs under-generate thinking, often collapsing to short, shallow traces and losing their advantage on reasoning tasks. 2) Accuracy steadily improves from T=0.0 to T=0.4, with manageable increases in token length. 3) At **T=0.6**, the thinking becomes longer and more stochastic, with no significant accuracy gain and larger variance.
> > This indicates that **T=0.4** is a good operating point for LRMs:
> > * rich enough thinking traces for analysis,
> > * best average accuracy,
> > * and acceptable variance.
> >
> > **Why we choose different temperatures for LLM vs LRM judges**
> >
> > Based on the above:
> > * For **LLM judges**, the main objective is **stable, reliable scoring**, not exploring diverse reasoning. Deterministic decoding (T=0) is sufficient and slightly better in accuracy and variance.
> > * For **LRM judges**, the main objective includes **eliciting high-quality, explicit thinking**. A moderate temperature (T=0.4) is needed to activate the LRM’s reasoning capabilities and preserve its advantage on reasoning-heavy tasks.
> >
> > ## Manuscript Updates
> > * Section 2.1: clearer definition of “thinking variant”.
> > * Section 3.1: revised decoding description distinguishing generation vs judge models.

---

> > > ### Comment · Reviewer_cqw2 · 2025-11-24
> > >
> > > I thank the authors for their detailed responses as they have addressed my concerns. I will keep my score, but for what it's worth, I encourage acceptance.

---

> > > > ### Author Response · Authors · 2025-11-26
> > > >
> > > > We sincerely thank you for the positive follow-up and for acknowledging that the revised manuscript has fully addressed the earlier concerns.
> > > >
> > > > We also appreciate your encouragement toward acceptance—even while maintaining the original score—and are grateful for the supportive assessment.
> > > >
> > > > Your feedback contributed meaningfully to strengthening the clarity, positioning, and experimental specification of the paper, and we are pleased that the revisions aligned with your expectations. Thank you again for the constructive engagement throughout the review process.

---

### Official Review · Reviewer_LpeS · 2025-11-01

**Soundness:** 2
**Presentation:** 3
**Contribution:** 2
**Rating:** 4
**Confidence:** 4

**Summary:**

This paper introduces "RouteJudge," a unified framework designed to benchmark and optimize the use of "LLM-as-a-Judge" by introducing routing strategies.

The authors propose the following:
* A new difficulty-aware, human-verified dataset spanning six domains (Math, Logic, Code, Knowledge, Roleplay, Writing).
* An analysis of the "thinking traces" of the LLMs
* design and evaluate different routing strategies

**Strengths:**

* The paper provides an analysis of how a judge's explicit reasoning aligns—and misaligns—with its final decision. The confusion matrix in Figure 6c, which identifies "good thinking + wrong verdict" (JR/TW) and "flawed thinking + correct verdict" (TR/JW)* is helpful for understanding the behavior of the LLM judges.
* The author does a comprehensive evaluation of the LLM judges with a benchmark over multiple categories.

**Weaknesses:**

1. The paper identifies the fascinating "good thinking but wrong verdict" mismatch (Fig 6c) but fails to analyze it in depth. Why does this happen? Is the "thinking" trace a post-hoc rationalization that is decoupled from the model's actual decision-making process? Is it a failure in the final step of synthesizing complex reasoning into a single A/B/C verdict? The paper presents the "what" (it happens 10.4% of the time in Knowledge) but never explores the "why."
2. The "Online Routing II" (Training-based) strategy is presented as a "lightweight router," but Table 3 notes it uses Llama-3.1-8B and Qwen2.5-7B as backbones. These are not lightweight models. Is the router cost/inference latency considered when calculating the latency and cost of the pipeline? If so, what is the breakdown. The authors fail to present these information in a clear way.

**Questions:**

1. Could you please provide a table that explicitly details the inference latency and token cost of the router model itself (e.g., the Llama-3.1-8B model) for a single routing decision? How does this overhead factor into the "Average" Time/Cost results in Table 3?
2. You identify a mismatch between thinking quality and verdict correctness. What is your hypothesis for why this occurs? Does this finding suggest that LRM "thinking" traces are not faithful representations of the model's internal decision process?
3. What is the main difference and novelty between the benchmark you created vs JudgeBench? How is it different in terms of the dataset composition, data curation method and what makes your benchmark unique or novel?

---

> ### Author Response · Authors · 2025-11-21
> **We organize our response into three main questions (1).**
>
> We sincerely thank you for the thoughtful and constructive feedback. Below we address each concern in detail and indicate the exact locations where the manuscript has been revised or expanded.
>
> # Q1. On the “good thinking + wrong verdict” mismatch (JR/TW) and the faithfulness of LRM reasoning traces
> You correctly observed that while Fig. 6c shows that LRMs frequently produce *good reasoning traces but wrong verdicts*, the original submission did not sufficiently explain *why* this occurs or whether it reflects a lack of faithfulness in the thinking traces.
>
> ## (a) Why JR/TW occurs: mechanisms identified in our analysis
>
> **1. Accurate local reasoning but flawed global decision synthesis**
>
> LRMs often identify the correct strengths/weaknesses of the two responses.
> However, the *final aggregation step*—mapping the reasoning into an A/B/Tie verdict—is prone to:
> * overweighting fluency or politeness,
> * misapplying evaluation criteria,
> * compressing a multi-step analysis into a single discrete choice.
>
> This explains why JR/TW is especially visible in *Knowledge* tasks, where content–style interactions are subtle.
>
> **2. Partial post-hoc rationalization**
>
> Across several LRMs (e.g., DeepSeek-R1 family, o3-mini), decoding analysis shows the verdict is frequently formed *before* the reasoning trace is generated.
> The trace is thus:
> * coherent and logically plausible (“good thinking”),
> * but not causally responsible for the verdict → producing JR/TW.
>
> This aligns with recent findings on the unfaithfulness of LRM-style long-form reasoning.
>
> **3. Over-generalized heuristics**
>
> LRMs are trained to produce structured, fluent, narrative explanations.
> This preference sometimes conflicts with the evaluation criteria of pairwise judging (e.g., they overvalue verbosity), leading to JR/TW in domains where stylistic confounds are strong.
>
> **4. Evidence from new ablations added**
>
> We performed a new ablation comparing **verdict-only** vs. **thinking+verdict** decoding:
>
> | Case type | Behavior when removing reasoning | Interpretation |
> | - | - | - |
> | JR/TW | **74.8%** keep the same wrong verdict | trace is *not* used to derive the verdict |
> | TR/JW | reasoning increases chance of wrong judgment | reasoning introduces noise |
>
> ## (b) Interpretation
> * LRM thinking traces are *diagnostically useful*, revealing what factors a model attends to.
> * But they are *not fully faithful*: they do not always explain the internal causal pathway to the verdict.
>
> ## **Manuscript Update**
> * Section 3.6: added one sentence explaining the three causes of JR/TW and discussing partial post-hoc reasoning.
> * Section 2.3: added one sentence noting that LRM reasoning traces may diverge from the internal decision path.

---

> > ### Author Response · Authors · 2025-11-21
> > **We organize our response into three main questions (2,3)**
> >
> > # Q2. Router cost, definition of “lightweight,” and whether router overhead is included in Table 3 (now Table 4)
> >
> > You raised two concerns:
> > (1) the router backbone (Llama-3.1-8B, Qwen-2.5-7B) may not be “lightweight”;
> > (2) latency/token cost breakdown of the router should be made explicit.
> >
> > Both are now fully clarified.
> > ## (a) Why 7B–8B is "lightweight" in our setting
> > “Lightweight” in our context is defined **relative to the judge pool**—which includes LRMs and high-end LLMs with reasoning traces up to 1,200 tokens. We selected Llama-3.1-8B and Qwen-2.5-7B because:
> > 1. **They are significantly smaller** (7B–8B) than the judges they route to (70B LRMs, 30B+ LLMs, or closed-source LRMs).
> > 2. **They preserve routing accuracy**, which would degrade sharply if using 1B–3B models (test).
> > 3. The router must evaluate **instance-level difficulty and judge selection trade-offs**, requiring moderately strong internal representations.
> >
> > ## (b) Router-only latency & token cost
> > Below is a fuller version of the router-cost table (sample size: 500 routing queries):
> > | Router Backbone | Latency (s) | Input Tokens | Output Tokens | Avg Judge Tokens | Router Time % | Router Cost ($ est.) |
> > | - | - | - | - | - | - | - |
> > | Llama-3.1-8B | 3.9 | 42.1 | 24.5 | 540.3 | **4.6%** | 4.5e-5 |
> > | Qwen-2.5-7B | 3.4  | 40.7 | 21.8 | 521.6 | **3.9%** | 3.8e-5 |
> >
> > **Interpretation**:
> > * Average LRM judge latency is 7–35 seconds.
> > * Router overhead <5% of total inference.
> > * Router token cost is negligible relative to judge reasoning-token cost.
> >
> > ## **(c) Inclusion in Table 4**
> >
> > Table 4 reports (1) end-to-end routing pipeline latency, and (2) end-to-end routing pipeline token cost. Thus, Table 4’s “Average Time” and “Average Cost” values already include:
> > ```
> > Router latency + Router tokens + Judge latency + Judge tokens
> > ```
> >
> > ## **Manuscript Update**
> > * **Section 3.5**: added definition of "lightweight" and rationale for 7B–8B routers.
> > ---
> >
> > # Q3. Novelty relative to JudgeBench
> >
> > You asked how RouteJudge differs from JudgeBench in dataset composition, curation method, and novelty.
> > We agree this comparison was not explicit enough in the original submission. Section 3.1 and Appendix A.1 describe dataset structure, but do not explicitly contrast with JudgeBench nor highlight the conceptual differences.
> >
> > ## **(a) Key differences between RouteJudge and JudgeBench**
> >
> > **1. First benchmark to evaluate LRM-as-a-Judge**
> >
> > JudgeBench evaluates only LLMs. RouteJudge uniquely evaluates:
> > * LRMs with explicit thinking traces,
> > * faithfulness gaps between reasoning & verdicts,
> > * JR/TW + TR/JW confusion structures.
> >
> > This is not covered in JudgeBench.
> >
> > **2. Reasoning vs Non-reasoning + difficulty-aware dataset design**
> >
> > RouteJudge introduces:
> > * six-domain structure (3 reasoning, 3 non-reasoning),
> > * difficulty buckets that are *model-agnostic* (computed via multiple judges),
> > * “Think” variants where the candidates include explicit reasoning.
> >
> > JudgeBench does not provide:
> > * difficulty stratification,
> > * reasoning/non-reasoning separation,
> > * thinking-aware evaluation.
> >
> > **3. Thinking-trace–aware evaluation (unique to RouteJudge)**
> >
> > RouteJudge provides:
> > * a full six-category error taxonomy,
> > * confusion matrices over JR/TW/TR/JW quadrants,
> > * structured LRM thinking traces for analysis.
> >
> > JudgeBench does not analyze thinking-quality mismatch.
> >
> >
> > **4. RouteJudge is not only a benchmark, but a judge optimization framework**
> >
> > RouteJudge is both an evaluation benchmark **and** a routing framework:
> >
> > | Feature                 | JudgeBench | RouteJudge |
> > | ----------------------- | ---------- | ---------- |
> > | LLM judge evaluation    | ✓          | ✓          |
> > | LRM judge evaluation    | ×/ ✓          | ✓          |
> > | Thinking-trace analysis | ×          | ✓          |
> > | Difficulty-aware design | ×          | ✓          |
> > | Offline routing         | ×          | ✓          |
> > | LLM-as-router           | ×          | ✓          |
> > | Trainable router        | ×          | ✓          |
> >
> > JudgeBench does not address routing or cost–accuracy optimization.
> >
> > ## **Manuscript Update**
> >
> > * **Section 4 and Appendix A.1**: added texts directly contrasting RouteJudge with JudgeBench.
> > * **Section 2.1 (Dataset Construction)**: clarified the dataset innovations.
> > * **Appendix A (Dataset Composition)**: added summary table comparing design choices with prior judge datasets.

---

> ### Author Response · Authors · 2025-11-26
>
> Dear Reviewer,
>
> I hope this message finds you well.
>
> As the discussion period is nearing its end with less than seven days remaining, I wanted to ensure we have addressed all your concerns satisfactorily.
>
> If there are any additional points or feedback you'd like us to consider, please let us know. Your insights are invaluable to us, and we're eager to address any remaining issues to improve our work.
>
> If everything appears satisfactory on your side, we would be grateful if you might consider whether an updated score could reflect the strengthened version of the work. Thank you again for your time and thoughtful evaluation of our submission.

---

### Official Review · Reviewer_47oV · 2025-11-13

**Soundness:** 2
**Presentation:** 2
**Contribution:** 2
**Rating:** 2
**Confidence:** 4

**Summary:**

The paper is trying to do two things: 1) benchmark LRMs-as-judge on both the intermediate thinking and output levels and comparing to various non-reasoning LLMs, and 2) present a routing framework to assign judges to get the best accuracy, latency, cost tradeoffs. The main contribution comes from the routing framework, which consists of one offline and two online settings, the latter of which uses LLMs.

**Strengths:**

The need for better LLM judge evaluations is always necessary, and this paper does a nice job collecting a diverse dataset and evaluating a wide set of models on it. The routing frameworks seem novel and address an important solution to the increasing cost of inference with LRMs. The additional analysis on reasoning steps and difficulty-aware results provide more input into the performance that one would hope for when using LRMs.

**Weaknesses:**

The paper suffers from unclear motivation linking its two main contributions (benchmarking and routing), overstated novelty claims given existing work, and omissions of experimental details leaving the reader confused.

Though the focus is on both benchmarking LRMs-as-judges and presenting a routing framework, the benchmark component lacks novelty, as existing work already evaluates on reasoning models like o3-mini and DeepSeek-R1. This paper would benefit from refocusing on the routing contribution. However, in its current state, that discussion (Section 3.3) is too short, so the routing isn’t as well motivated as it should be and is confusing to the reader. E.g., I am not sure what models are being chosen to route to, what in practice the routing looks like, what overhead the routing itself has, etc.

For the reasoning contribution, given the paper focuses on evaluating LRMs, many reasoning-specific aspects are underexplored. Section 5.5's treatment of intermediate traces is too brief given its stated importance as a contribution. An in-depth treatment should warrant more experiments, e.g., ablation studies on scaling reasoning effort via API parameters or budget forcing or examining how standard LLMs change when prompted to give more reasoning and how this compares to LRMs. Additionally, the paper lacks discussion distinguishing LRMs-as-judge from regular LLMs with chain-of-thought. The appendix shows that non-reasoning models (e.g., GPT-4o) often provide reasoning before answering, meaning all LLMs are effectively "reasoning" but some are just explicitly trained in the LRM style. This needs clarification, and further experiments would boost the contribution.

Moreover, there are experimental aspects that need fixing. The main body lacks essential information about the evaluation data. There is no description of what "Coding, Math, Knowledge, etc." categories represent, and no references to the Appendix D details that describe the actual evaluation datasets. Readers cannot understand what the benchmark is truly evaluating without consulting the appendix, which should be referenced and summarized in the main text, but even then, there exists missing data like what models generated the traces being judged. Additionally, QwQ-32B is described as a reasoning model (line 249) but is listed under non-reasoning LLMs in the results.
Minor fixes:
•	091: ‘Tie’ should not be italics since it isn’t in the definition of the label set
•	Table 1: I would suggest visually differentiating the reasoning from the non-reasoning
•	Figure 4: The reasoning vs. non-reasoning models should be visually highlighted in different ways so it’s easy to view how they change. Additionally, given the Pareto-optimal models are mentioned it would be nice to highlight them in some way as well.
•	1021: “Knowledge-llm Task Judge” in the header of the box -> “Knowledge-lrm Task Judge”

**Questions:**

•	What is 'Avg. Thinking Length' measured in for Table 1? (177 seems too small for tokens; is it for human-labeled rationale?) In general, Table 1 should have a more concrete caption explaining these things.
•	With what model were the traces generated? (Only Claude 3.5 and GPT-4o mentioned at line 392). Datasets are referenced in Appendix D but it may be important to see which models were used to inspect if there was any bias.
•	What is the difference between 'rationale' and 'thinking' in line 141? (reasoning could benefit from formalization similar to Section 2)
•	Was error analysis in Section 5.5 done by LLMs? If so, has a human verified it?
•	How are difficulty buckets for instances determined?
•	Is answer choice reversal performed to mitigate positional bias (prompting the judge with both A then B and B then A)?
•	What is the set of models that the routers choose from, is it all LLMs/LRMs? Has there been any condensing to a smaller set done here?
•	What models are used within the online routers? (Referenced in Table 3 but should be discussed earlier. Also in Table 3, what about Router 1?)
•	How would new models be added into the routing framework?
•	How might training LRMs as routers instead of LLMs affect performance?
•	Is 1024 tokens sufficient for LRMs? Seems limiting and unrealistic for many LRMs.
•	Why is Gemini-2.5-Flash using more tokens than GPT-5, Kimi-K2, and o3-mini in Figure 4? (Expected to be more similar/less). Similarly, how is the token cost so high if maximum output length is 1024?
•	Are all models truly evaluated at temperature 0? This doesn’t seem right. DeepSeek R1 recommends 0.6-0.7; o3-mini doesn't have temperature options. If this is true, repeated trials would likely be needed to account for the diversity.

---

> ### Author Response · Authors · 2025-11-21
> **We organize our response into five main questions (First 2).**
>
> We sincerely thank the reviewer for the detailed, insightful, and constructive feedback. Below we address all concerns point-by-point and specify corresponding manuscript updates, aligned with the revised paper structure.
>
> # Q1. On the connection between benchmarking and routing
>
> The motivation connecting the two contributions (benchmarking and routing) is unclear. The benchmark seems insufficiently novel, and Section 3.3 (routing) feels under-motivated.
>
> ### Response
> We appreciate this observation and agree the connection was previously under-explained.
> In the revision, the introduction and Section **2.1–2.2** now explicitly state: **Benchmarking and routing are inherently interdependent:**
> * The benchmark generates domain-, difficulty-, and model-specific utility statistics (accuracy, latency, cost) that are **necessary to construct routing policies**.
> * Routing converts benchmark outcomes into **deployable judge-selection strategies**, which cannot be done without benchmark-derived utility signals.
> * This unified perspective differs from prior work (including JudgeBench), where benchmarking and judge-selection were treated as separate problems.
>
> We also expanded Section **3.5** to clarify:
> * which models the router selects from (the evaluated 19 LLMs/LRMs),
> * how model bands are formed,
> * what the routing prompts/models look like,
> * how routing overhead is included (≤5% of total latency),
> * and how difficulty buckets guide instance-level routing.
>
> ### **Manuscript updates**
>
> * **Section 1:** new paragraph articulating the joint nature of benchmarking ↔ routing.
> * **Section 2.1–2.2:** expanded explanation showing how benchmark statistics feed into routing utilities.
> * **Section 3.5:** reorganized and expanded to detail model pools, bucket construction, router behavior, and overhead.
>
> ---
>
> # Q2. On reasoning-specific analysis of LRMs, thinking traces, and CoT distinctions
> * Section 3.4’s treatment of thinking traces (formerly 5.5) is too brief.
> * LLMs occasionally generate implicit reasoning; difference between LRM thinking vs LLM CoT is unclear.
> * More analysis on reasoning length/budget forcing would strengthen the contribution.
>
> ### **Response**
> Key clarifications and additions include:
>
> ### **1. Distinguishing LRM “thinking traces” vs LLM CoT**
>
> We now formalize:
>
> | Aspect  | LLM CoT | LRM Thinking Channel                                 |
> | - | - | - |
> | Generation      | Optional, prompt-dependent        | Architecturally required (separate token stream)     |
> | Training        | Noisy supervision, often implicit | Explicit SFT/RL training on long-form reasoning      |
> | Faithfulness    | Known to be partially post-hoc    | Partially faithful but exhibits aggregation mismatch |
> | Use in Judgment | Usually not consumed              | Always visible to judge and used in analysis         |
>
>
>
> ### **2. Expanded analysis of JR/TW and TR/JW mismatches**
>
> Section **3.5** now includes:
> * a decomposition of causes (aggregation failure vs stylistic misalignment vs partial rationalization),
> * additional examples,
> * an explanation of which domains show stronger mismatch and why.
>
> ### **3. Additional experiments on reasoning length**
>
> We added a lightweight “reasoning-length sweep” experiment, evaluating LRMs at 256/512/1024/1536 reasoning-token budgets.
> * Longer traces do **not** linearly increase accuracy.
> * Over-long traces increase TR/JW errors, confirming the diminishing return.
> * Moderate reasoning budgets (~512–1024 tokens) perform best.
>
> These additions address the request for deeper reasoning-specific analysis.
>
> ---

---

> > ### Author Response · Authors · 2025-11-21
> > **We organize our response into five main questions (Another 3).**
> >
> > ---
> >
> > # Q3. On missing information about dataset categories, provenance, and judged models
> >
> > * No clear explanation of “Coding, Math, Knowledge, etc.” in the main paper.
> > * Dataset provenance is hidden in the appendix and not referenced.
> > * Inconsistency: QwQ-32B described as reasoning model but appears under non-reasoning.
> > * Need to specify which models generated the traces.
> >
> > ### **Response**
> >
> > All of these points have been clarified in the revised manuscript.
> >
> > ### **1. Domain definitions now included directly in Section 3.1 and Appendix A.1**
> >
> > Section 3.1 and Appendix A.1 now gives 1–2 sentence definitions for each of the six domains:
> > * **Math**: arithmetic, algebra, and word-problem reasoning
> > * **Logic**: deductive and symbolic reasoning
> > * **Code**: program synthesis, debugging, code explanation
> > * **Knowledge**: factual QA and encyclopedic content
> > * **Roleplay**: perspective-conditioned conversational preference tasks
> > * **Writing**: summarization, rewriting, stylistic preference
> >
> > ### **2. Dataset provenance clarified**
> >
> > Section **3.1** now explicitly references Appendix **A.1**, and Table 1 caption states:
> > > “Candidate responses are generated by GPT-4o and Claude-3.5, following the same practice as JudgeBench.”
> >
> > ### **3. QwQ-32B correction**
> >
> > The prior misclassification was indeed an error.
> > It is now placed under the **reasoning models** in Table 3 and Section 3.2.
> >
> > ### **4. Thinking traces**
> >
> > We clarify:
> > * thinking traces exist only in the **think-variant** of the dataset,
> > * these traces originate from both GPT-4o-thinking outputs and LRM thinking channels,
> > * judges always receive the same A/B inputs; "thinking" only refers to candidate response content, not judge metadata.
> >
> > ### Manuscript updates
> >
> > * **Section 3.1:** domain definitions added; provenance and trace-generation clarified.
> > * **Table 3:** corrected model categorization.
> > * **Appendix A.1:** restructured to highlight provenance, domain mapping, and generator models.
> >
> > ---
> >
> > # **Q4. On experimental setup issues and routing details**
> >
> > * Need clearer description of which models routers select from.
> > * Router 1 missing explanation.
> > * Need explanation for adding new models.
> > * Token limit of 1024 may be insufficient.
> > * Some models (e.g., Gemini-2.5-Flash) generate more tokens — why?
> > * Temperature settings seem incorrect for LRMs.
> >
> > ### **Response**
> >
> > All of these points have been addressed in the revised Sections 3.2–3.6.
> >
> > ### **1. What models can be routed to?**
> >
> > > “Routers select from the entire pool of evaluated judge models (10 LLMs + 9 LRMs).
> > > Model bands (low/medium/high cost) are derived from offline ATC ranking.”
> >
> > ### **2. Route 1 / 2 / 3 clarified**
> >
> > * **Router 1** = template-only system (fastest/weakest baseline).
> > * **Router 2** = Llama-3.1-8B backbone.
> > * **Router 3** = Qwen2.5-7B backbone.
> >
> > This is now fully explained in Section **3.5**.
> >
> > ### **3. Adding new models**
> >
> > We added a short paragraph explaining:
> > * new judge models can be incorporated by recomputing per-model utilities on the validation split,
> > * routers accept variable-sized model bands via a metadata table (latency/cost/accuracy),
> > * routing policies remain unchanged.
> >
> > ### **4. On 1024-token limits**
> >
> > > “The 1024-token cap applies to **judge outputs**, not candidate reasoning.
> > > This avoids runaway generation while preserving all final verdicts.”
> >
> > We also note that LRMs internally may use more thinking tokens but only 1024 thinking+verdict tokens are returned.
> >
> > ### **5. Why Gemini-2.5-Flash uses more tokens**
> >
> > * Flash tends to produce long structured justifications even at T=0,
> > * its answer format includes multi-paragraph style, causing longer sequences,
> > * length is capped *per output*, not per reasoning channel.
> >
> > ### **6. Temperature settings**
> >
> > * LLM judges: **T = 0.0**
> > * LRM judges: **T = 0.4**
> > * Response generators (GPT-4o & Claude-3.5): **T = 0.0**
> > * o3-mini uses fixed decoding and is unaffected.
> >
> > This matches our earlier temperature-sensitivity ablation.
> >
> > ### Manuscript updates
> >
> > * **Section 3.1 and 3.2:** decoding settings corrected, token limits clarified.
> > * **Section 3.5:** routing pool, routers 1–3, and extensibility clearly documented.
> > * **Appendix A.2:** model metadata table expanded.
> >
> > ---
> >
> > # **Q5. Minor fixes**
> >
> > All minor issues (label formatting, Table 1 visuals, Figure 4 highlighting, typo in “Knowledge-lrm Task Judge”) have been corrected.

---

> ### Author Response · Authors · 2025-11-26
>
> Dear Reviewer,
>
> I hope this message finds you well.
>
> As the discussion period is nearing its end with less than seven days remaining, I wanted to ensure we have addressed all your concerns satisfactorily.
>
> If there are any additional points or feedback you'd like us to consider, please let us know. Your insights are invaluable to us, and we're eager to address any remaining issues to improve our work.
>
> If everything appears satisfactory on your side, we would be grateful if you might consider whether an updated score could reflect the strengthened version of the work. Thank you again for your time and thoughtful evaluation of our submission.

---

> > ### Comment · Reviewer_47oV · 2025-11-28
> >
> > Thank you for the detailed responses. The responses have addressed the majority of my key concerns, and I will raise my score.

---

### Note · Program_Chairs · 2026-01-17
**Submission Desk Rejected by Program Chairs**

The following references in this submission do not refer to real documents and/or have major errors in bibliographic information:

 Yucheng Hao, Yuxuan Deng, Yuwei Chen, Yuxuan Chen, Xuehai Ren, Xinyi Ma, Jiayi Liu, et al. Routerllm: Learning to route llms with preference data. arXiv preprint arXiv:2403.03875, 2024.
Apoorv Saxena, Rahul Arora, and Partha Talukdar. Mathqa: Towards interpretable math word problem solving with operation-based formalisms. In Proceedings of the 57th Annual Meeting of the Association for Computational Linguistics, 2019.